# Genetic transformation of structural and functional circuitry rewires the *Drosophila* brain

**Sonia Sen[1,2], Deshou Cao[3], Ramveer Choudhary[1,4], Silvia Biagini[1,4], Jing W Wang[3], Heinrich Reichert[5], K VijayRaghavan[1]\***

[1]Department of Developmental Biology and Genetics, National Centre for Biological Sciences, Tata Institute for Fundamental Research, Bangalore, India; [2]Manipal University, Manipal, India; [3]Division of Biological Sciences, University of California, San Diego, San Diego, United States; [4]Institute for Molecular Oncology, Milan, Italy; [5]Department of Biozentrum, University of Basel, Basel, Switzerland

**Abstract** Acquisition of distinct neuronal identities during development is critical for the assembly of diverse functional neural circuits in the brain. In both vertebrates and invertebrates, intrinsic determinants are thought to act in neural progenitors to specify their identity and the identity of their neuronal progeny. However, the extent to which individual factors can contribute to this is poorly understood. We investigate the role of *orthodenticle* in the specification of an identified neuroblast (neuronal progenitor) lineage in the *Drosophila* brain. Loss of *orthodenticle* from this neuroblast affects molecular properties, neuroanatomical features, and functional inputs of progeny neurons, such that an entire central complex lineage transforms into a functional olfactory projection neuron lineage. This ability to change functional macrocircuitry of the brain through changes in gene expression in a single neuroblast reveals a surprising capacity for novel circuit formation in the brain and provides a paradigm for large-scale evolutionary modification of circuitry.

**\*For correspondence:** vijay@ncbs.res.in

## Introduction

Animals display a wide repertoire of complex and adaptable behaviours executed by equally complex nervous systems. Understanding how the vast number of diverse cell types is assembled into functional neural circuits in complex brains during development is a major challenge. Studies of lineage tracing and circuit mapping reveal that heterogeneous pools of neural progenitors sequentially generate series of neuronal progeny, and that such lineally related neurons with shared developmental histories often share functional connectivity in the brain. In consequence, neural lineages can be considered to form neuroanatomical units of projection that represent the developmental basis of the functional circuitry of the brain (*Pearson and Doe, 2004*; *Cardona et al., 2010*; *Pereanu et al., 2010*; *Custo Greig et al., 2013*; *Franco and Muller, 2013*; *Gao et al., 2013*; *Kohwi and Doe, 2013*).

This is exemplified in *Drosophila* where the tens of thousands of neurons that comprise the adult brain are generated during development by a set of approximately 100 pairs of individually identifiable neural stem cells called neuroblasts (*Truman and Bate, 1988*; *Urbach et al., 2003*; *Urbach and Technau, 2003a, 2003b*; *Technau et al., 2006*). Each neuroblast gives rise to a specific, invariant lineage of post-mitotic neural cells in a highly stereotyped manner and in many cases, lineally related neurons share functional connectivity and many neuroanatomical features such as innervation of common neuropiles in the brain and common axon tract projection patterns (*Pereanu and Hartenstein, 2006*; *Ito et al., 2013*; *Lovick et al., 2013*; *Wong et al., 2013*; *Yu et al., 2013*). Examples of this are the four neuroblast lineages that give rise to the intrinsic cells of the mushroom body, a neuropile

**eLife digest** The cells in the brain—including the neurons that transmit information—work together in groups called neural circuits. These cells develop from precursor cells called neuroblasts. Each neuroblast can produce many cells, and it is likely that cells that develop from the same neuroblast work together in the adult brain in the same neural circuit. How the adult cells develop into their final form plays an important role in creating a neural circuit, but this process is not fully understood.

In many animals, the complexity of their brain makes it difficult to follow how each individual neuroblast develops. However, all of the neuroblasts in the relatively simple brain of the fruit fly *Drosophila* have been identified. Furthermore, the genes responsible for establishing the initial identity of each neuroblast in the *Drosophila* brain are known. These genes may also determine which adult neurons develop from the neuroblast, and when each type of neuron is produced. However, the extent to which a single gene can influence the identity of neurons is unclear.

Sen et al. focused on two types of neuroblasts, each of which, although found next to each other in the developing *Drosophila* brain, produces neurons for different neural circuits. One of the neuroblasts generates the olfactory neurons responsible for detecting smells; the other innervates the 'central complex' that has a number of roles, including controlling the fly's movements. A gene called *orthodenticle* is expressed by the central complex neuroblast, but not by the olfactory neuroblast, and helps to separate the two neural circuits into different regions of the fly brain.

Sen et al. found that deleting the *orthodenticle* gene from the central complex neuroblast causes it to develop into olfactory neurons instead of central complex neurons. Tests showed that the modified neurons are completely transformed; they not only work like olfactory neurons, but they also have the same structure and molecular properties. Sen et al. have therefore demonstrated that it is possible to drastically alter the circuitry of the fruit fly brain by changing how one gene is expressed in one neuroblast. This suggests that new neural circuits can form relatively easily, and so could help us to understand how different brain structures and neural circuits evolved.

compartment involved in learning and memory, or the five neuroblast lineages that innervate the antennal lobe, the primary olfactory processing centre in the fly brain (*Ito and Hotta, 1992*; *Ito et al., 1997*; *Stocker et al., 1997*; *Jefferis et al., 2001*; *Das et al., 2008, 2011, 2013*; *Lai et al., 2008*; *Chou et al., 2010*). Thus, neuroblast lineages are considered to form neuroanatomical units of projection that represent the developmental basis of the functional macrocircuitry of the fly brain (*Pereanu and Hartenstein, 2006*; *Cardona et al., 2010*; *Lovick et al., 2013*; *Wong et al., 2013*).

Comparable principles are manifested in the developing cerebral cortex of vertebrates, which consists of diverse neurons organized into six distinct layers, each of which is laid in place sequentially during development. Neural progenitors in the cortex are known to be multipotent, capable of generating neurons that populate each of the layers. Lineage tracing experiments in mice suggest that lineally related neurons occupy columns spanning across the layers of the cerebral cortex, as proposed in the 'radial unit hypothesis' (*Rakic, 1988*; *Yu et al., 2009b*; *Li et al., 2012*; *Ohtsuki et al., 2012*). Furthermore, lineally related neurons also show a propensity to interconnect and have functional similarity, for example, similar orientation preferences in the visual cortex (*Li et al., 2012*; *Ohtsuki et al., 2012*). Thus in vertebrates as in invertebrates, developmental history and lineage relationships govern the assembly of functional circuits.

In order to understand how lineally specified circuits develop in the brain, it is critical to understand the molecular mechanisms that confer unique identities to neural progenitors and their lineages. Studies on the molecular genetics of brain development indicate that neural progenitors emerge from the embryonic neuroectoderm where unique spatial information represented by unique combinations of gene expression specifies unique identities to the progenitors. For example, in *Drosophila*, the embryonic neuroectoderm becomes spatially regionalized due to the action of embryonic patterning genes, which define the anterior-posterior and dorsal-ventral body axes. Their combined expression creates a Cartesian coordinate-like gene expression system in the neuroectoderm, resulting in unique domains of expression of developmental control genes along the neuroectoderm (*Skeath and Thor, 2003*; *Urbach and Technau, 2004*; *Technau et al., 2006*). Changes in the combinatorial expression pattern of these genes in specific domains can lead to changes in the identities of the

neuroblasts that delaminate from the neuroectoderm during embryogenesis (e.g., *Deshpande et al., 2001*). The process of spatial regionalization of the embryonic neuroectoderm is very similar in vertebrates. Homologous embryonic patterning genes result in unique domains of combinatorial gene expression along the vertebrate neuroectoderm (*Reichert and Simeone, 2001*; *Lichtneckert and Reichert, 2005*, *2008*; *Reichert, 2009*). Thus, in both vertebrates and invertebrates, the cells of the neuroectoderm acquire unique spatial information in the form of a combinatorial code of gene expression, which is conferred by embryonic patterning genes. It is noteworthy that neural progenitors also use temporal information (typically a series of sequentially expressed transcription factors) to generate neuronal diversity within lineages (*Pearson and Doe, 2004*; *Lin and Lee, 2012*; *Kohwi and Doe, 2013*). While spatial cues convert a homogenous pool of progenitors into heterogeneous populations, temporal cues result in the ordered production of different neural subtypes from each progenitor.

Given that spatial information in the neuroectoderm, in the form of embryonic patterning genes, imparts heterogeneity to neural progenitor populations, it is likely that these genes might also act as intrinsic determinants in the progenitor to give lineages their unique identities and hence determine their place in neural circuitry. This implies that spatially encoded intrinsic factors determine the identity of the progenitor and, as a consequence, the unique circuit features of its neural lineage. According to this, removal or addition of one or more of these genes could lead to a change in neuroblast identity resulting in transformation of the neuronal lineage and the lineal circuitry that derives from it. However, the extent to which individual transcription factors can contribute to this specification of neuroblast identity is not well understood.

In order to test this, it is important to be able to uniquely identify individual neural progenitors and their lineages in the brain. The complexity of the vertebrate brain makes it difficult to conduct such an analysis at the resolution of single progenitors and single lineages. However, each of the neuroblasts in the *Drosophila* brain has been identified and their lineages characterized in the larval and adult brains (*Ito et al., 2013*; *Lovick et al., 2013*; *Wong et al., 2013*; *Yu et al., 2013*). Furthermore, each of these neuroblasts has also been characterized by the expression of a specific combination of spatial genes, which could act as cell intrinsic determinants in the specification of unique neuroblast identity and hence control lineage-specific neuronal cell fate (*Skeath and Thor, 2003*; *Urbach and Technau, 2004*; *Technau et al., 2006*). This allows an investigation of the role of putative intrinsic determinants by changing their expression in identified stem cells and assessing its effect at the lineage level in an otherwise normal brain.

Here, we focus on two identified neuroblast lineages in the *Drosophila* brain, LALv1 and ALad1, which develop in close spatial proximity to each other in the larval brain but become spatially segregated in the adult brain. While the ALad1 neuroblast generates olfactory projection interneurons that innervate the antennal lobe, the LALv1 neuroblast generates wide-field interneurons that innervate the central complex. We show that *orthodenticle* (*otd*), an embryonic patterning gene involved in specifying the anterior-most regions of the neuroectoderm and embryonic brain (*Lichtneckert and Reichert, 2008*; *Reichert and Bello, 2010*), is expressed during development in the LALv1 neuroblast lineage but not in the ALad1 neuroblast lineage. Remarkably, loss of *otd* from the LALv1 neuroblast results in a complete transformation in the identity of the neurons that derive from this lineage. The *otd* null LALv1 neurons transform into antennal lobe projection interneurons similar to the ALad1 lineage, and this transformation includes a complete change in the neuroanatomy of the neurons, a change in their molecular properties as well as in their functional connectivity. This remarkably complete respecification of a neuroblast lineage upon the mutation of a single gene in the brain demonstrates that intrinsic determinants acting in the neuroblast during development specify the identity of its neural progeny and the macrocircuitry that these progeny establish. This large-scale modification of functional circuits in the brain by a single transcription factor in a single stem cell is unprecedented and reveals a surprising capacity for novel neural circuit formation in the developing brain, which may provide a paradigm for large-scale evolutionary modification of brain connectivity.

## Results

### Development, morphogenesis, and differential expression of Otd in two identified central brain neuroblast lineages, LALv1 and ALad1

We focused our analysis on two identified neuroblast lineages referred to as LALv1 and ALad1 (*Pereanu and Hartenstein, 2006*; *Lovick et al., 2013*) (see 'Materials and methods' for lineage nomenclature).

During postembryonic development in the larval brain, the adult-specific (postembryonically generated) neural progeny of these lineages have their cell bodies clustered close to each other, dorsal to the larval antennal lobe (*Figure 1A,B*). Although their cell body clusters are closely apposed, the two lineages can be easily identified based on their distinct and unique axon tracts that project to different brain regions (*Pereanu et al., 2010*; *Das et al., 2013*; *Lovick et al., 2013*).

The anatomical features of these two wild-type lineages can be visualized by MARCM clonal labelling (randomly induced neuroblast clones; ubiquitous *Tub-Gal4* driver). The ALad1 lineage initially projects its axon tract medially, dorsal to the larval antennal lobe, then turns posteriorly and projects towards the protocerebrum via the medial antennal lobe tract (*Figure 1B,C*) (*Das et al., 2013*; *Lovick et al., 2013*). The LALv1 lineage initially projects its axon tract ventro-medially, posterior to the larval antennal lobe, then loops dorsally and splits into two secondary axon tracts (*Figure 1B,D*) (*Spindler and Hartenstein, 2011*; *Lovick et al., 2013*).

In addition to the differences in axonal trajectories, we found that these two lineages also differed in their expression of the transcription factor Otd. Co-immunolabelling for the homeodomain transcription factor Otd and for Neurotactin (to identify lineage-specific axon tracts) shows that the LALv1 neuroblast (*Figure 1F*) and all of its lineal progeny (white dotted lines in *Figure 1F* and inset in *Figure 1D*) express Otd. In contrast, neither the ALad1 neuroblast nor its lineal progeny are found to express Otd (*Figure 1I–L* and inset in *Figure 1C*).

In the adult brain, the neural progeny of the ALad1 lineage are olfactory projection neurons, which innervate the glomeruli of the antennal lobe and the neural progeny of the LALv1 lineage are widefield interneurons that innervate the central complex, a sensorimotor integration centre in the fly brain (*Ito et al., 2013*; *Wong et al., 2013*; *Yu et al., 2013*). To study the neuroanatomical features of the two lineages in the mature brain, we took advantage of the fact that they are differentially labelled by four enhancer-Gal4 driver lines in the adult brain. Thus, the adult LALv1 lineage is labelled by the *OK371*-Gal4 (a glutamatergic neuron label) and *Per*-Gal4 driver lines (*Figure 1O*, *Table 1*), while the ALad1 lineage is not. Conversely, the adult ALad1 lineage is labelled by the *Cha7.4*-Gal4 (a cholinergic neuron label) and *GH146*-Gal4 lines, while the adult LALv1 lineage is not (*Figure 1P*, *Table 1*).

MARCM clonal labelling of the neurons in the adult ALad1 lineage using *GH146-Gal4* or *Cha7.4*-Gal4 drivers shows that their cell bodies are positioned dorsal to the adult antennal lobe, their dendrites innervate the antennal lobe glomeruli and their axons exit the lobe via the medial antennal lobe tract (*Figure 1M,N,P*). The axons then project dorso-posteriorly to innervate the calyx of the mushroom body and the lateral horn (*Video 1*) (*Ito et al., 2013*; *Wong et al., 2013*; *Yu et al., 2013*).

MARCM clonal labelling of the neurons in the adult LALv1 lineage using *Per-Gal4* or *OK371-Gal4* drivers shows that their cell bodies are positioned ventral to the adult antennal lobe and their axons project into the loVM tract, which courses posteriorly behind the adult antennal lobe, then loops dorsally creating the prominent LEp fascicle, which innervates the central complex neuropiles and the lateral accessory lobe (*Video 2*) (*Figure 1O*) (*Spindler and Hartenstein, 2011*; *Ito et al., 2013*; *Lovick et al., 2013*; *Wong et al., 2013*; *Yu et al., 2013*). As in the corresponding larval lineages, the adult LALv1 neuroblast lineage expresses Otd while the adult ALad1 neuroblast lineage does not (insets in *Figure 1O,P*). It is noteworthy that in the adult brain the cell bodies of the LALv1 neurons are located ventral to the adult antennal lobe, whereas in the larval brain the position of the cell bodies is dorsal to the larval antennal lobe (compare *Figure 1A,M*). This change in cell body position of the LALv1 lineage occurs as a consequence of the morphogenetic changes associated with the de novo development of the adult antennal lobe (*Jefferis et al., 2004*; *Spindler and Hartenstein, 2011*; *Lovick et al., 2013*; *Wong et al., 2013*).

In summary, the LALv1 neuroblast and its progeny, which innervate the central complex, express *otd* throughout brain development as well as in the adult brain, while the ALad1 neuroblast and its progeny neurons, which innervate the antennal lobe, do not.

## Loss of Otd from the LALv1 neuroblast results in lineage identity transformation

As *otd* is expressed in all of the cells of the central complex lineage—the neuroblast and the postmitotic neurons—we tested its possible function in both these cell types. In order to do this, we used MARCM-based clonal mutational methods to generate GFP-labelled *otd^{-/-}* clones in the LALv1 lineage in an otherwise heterozygous background (*Lee and Luo, 2001*). Using this technique, it is possible to genetically inactivate *otd* in the postmitotic neurons, the GMC, or the neuroblast, thus allowing

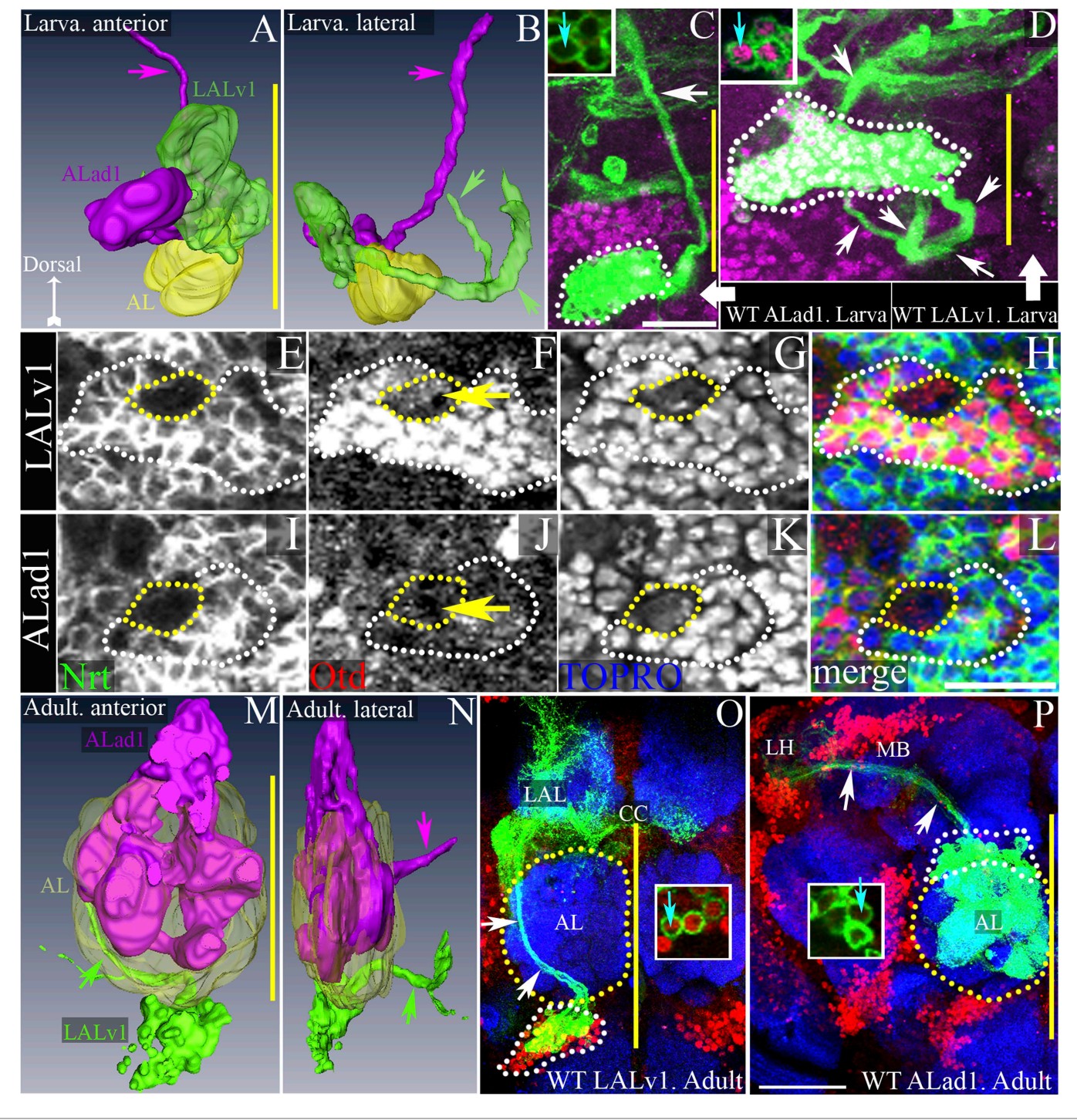

**Figure 1**. Development, morphogenesis, and differential Otd expression in two identified central brain neuroblast lineages, LALv1 and ALad1. (**A** and **B**) show anterior and lateral views of 3D reconstructions of the LALv1 (green) and the ALad1 (magenta) lineages in the larval brain. (**A**) shows that the cell bodies are closely apposed to each other and lie above the larval antennal lobe (AL, yellow), (**B**) shows their tracts diverge—the ALad1 tract (magenta) projects dorsally and the LALv1 tract (green) projects posteriorly behind the AL and splits. (**C** and **D**) show WT MARCM clones of the larval ALad1 and LALv1 lineages, respectively. Their cell bodies are outlined by white dotted lines and the white arrows trace their tracts. Insets in **C** and **D** show that while LALv1 cells (cyan arrow in **D**) express Otd, ALad1 cells (cyan arrow in **C**) do not. (**E–L**) is a third larval instar brain (CS) immunolabelled with neurotactin (green, to identify lineages), Otd (red) and TOPRO-3 (to label nuclei). The LALv1 lineage is documented in **E–H**, and the ALad1 lineage is documented in **I–L**. The neuroblasts are marked with yellow dotted lines and the lineages are marked with white dotted lines. The LALv1 neuroblast expresses Otd

*Figure 1. Continued on next page*

*Figure 1. Continued*

(yellow arrow in **F**) and the ALad1 neuroblast does not (yellow arrow in **J**). (**M** and **N**) show anterior and lateral views of 3D reconstructions of the LALv1 (green) and the ALad1 (magenta) lineages in the adult brain. Note that the adult antennal lobe (AL, yellow in **M** and **N**) is situated between the ALad1 lineage (antero-dorsal to AL) and the LALv1 lineage (ventral to AL) and the cell bodies of these lineages are not closely apposed anymore. The arrows in **M** and **N** indicate the ALad1 tract (magenta), which projects dorsally towards the protocerebrum and the LALv1 tract (green), which projects posterior to the AL. (**O** and **P**) show WT clones of the adult LALv1 and ALad1 lineages, respectively. Their cell bodies are outlined by white dotted lines and the AL is outlined by yellow dotted lines. White arrows trace the tracts of these lineages. The LALv1 lineage innervates the lateral accessory lobe (LAL) and the central complex (CC). The ALad1 lineage innervates the calyx of the mushroom body (MB) and lateral horn (LH). The midline is represented by a yellow line in all images. Scale bars in **C** (applicable to **D**) and in **L** (applicable to **E**–**L**) are 20 μm. Scale bar in **P** (applicable to **O**) is 50 μm. Genotypes in **C** and **D**: *FRT19A/FRT19A,Tub-Gal80,hsFLP; Tub-Gal4,UAS-mCD8::GFP/+*. Genotype in **O**: *FRT19A/FRT19A,Tub-Gal80,hsFLP; Per-Gal4,UAS-mCD8::GFP/+*. Genotype in **P**: *FRT19A/FRT19A,Tub-Gal80,hsFLP; GH146-Gal4,UAS-mCD8::GFP/+*.

us to assess its role in each of these cell types (see schematic in *Figure 2A*). We generated such *otd*⁻/⁻ clones early in larval development and analyzed them in the adult brain.

Using this technique, we first investigated a possible requirement of *otd* in the postmitotic neurons of the central complex lineage. In these experiments, in which we used the *OK371-Gal4* and *Per-Gal4* driver lines to label the MARCM clones, we obtained a total of seven wild-type single cell clones and 11 *otd*⁻/⁻ single cell clones. Although we have not dated the birth of these clones precisely (matched the time of clone generation), the wild-type single cell clones that we obtained in our experiments were very similar to those described previously (*Yu et al., 2009a*). Six of these single cell wild-type clones consisted of neurons that innervated both the lateral accessory lobe as well as one of the noduli of the central complex (*Figure 2B*) and one clone only innervated the lateral accessory lobe (*Figure 2C*). All of the 11 *otd*⁻/⁻ single cell clones we recovered also displayed a similar neuroanatomy. Their cell bodies were located ventral to the antennal lobe, their axons coursed through the loVM and LEp tracts and they all innervated the lateral accessory lobe and one of the noduli of the central complex (*Figure 2G,H*). Thus, loss of *otd* function from the postmitotic neurons did not result in any gross defects in the neuroanatomy of these neurons. It is however possible, that there are fine-scale changes in the arborisation of these neurons within the lateral accessory lobe and the central complex that we were unable to identify. It is also possible that *otd* function is required in the GMC for the targeting of the postmitotic neurons (see schematic in *Figure 2A*). However, in our experiments, we never obtained two cell GMC clones in order to be able to address this possibility.

We then asked if *otd* might be required in the neuroblast of the LALv1 lineage for its proper development. In order to test this, we inactivated *otd* in the neuroblast during early larval development and analyzed the neuroanatomy of the resultant labelled wild-type and *otd*⁻/⁻ mutant neurons in the adult brain. In these experiments, in which we used the *OK371-Gal4* and *Per-Gal4 driver* lines to label the MARCM clones, we recovered 6 and 14 wild-type clones, respectively. As expected, all the wild-type neuroblast clones displayed the neuroanatomy of the central complex lineage as described above (ventrally position cell bodies, axon projection via the loVM and LEp tracts and innervations in the lateral accessory lobe and central complex. *Figure 3B–E,J–M*). However, when we generated *otd*⁻/⁻ neuroblast clones in the LALv1 lineage (identifiable by the loss of Otd staining in the corresponding cell cluster ventral to the antennal lobe; white arrowhead in *Figure 3G,O*), neither of these drivers labelled the mutant LALv1 lineage (*Figure 3F–I,N–Q*).

In order to investigate the neuroanatomy of the *otd*⁻/⁻ LALv1 lineage further, we utilized the ubiquitously expressed *Tub-Gal4* driver to label neuroblast clones and recovered 19 WT and 37 *otd*⁻/⁻ neuroblast MARCM clones in the LALv1 lineage. While the wild-type neurons displayed all the features of the LALv1 lineage described above (*Figure 4—figure supplement 1A*), the

**Table 1.** Summary of the specific molecular changes in the LALv1 and Alad1 lineages

| | WT LALv1 | *otd*⁻/⁻ LALv1 | WT ALad1 |
|---|---|---|---|
| OK371-Gal4 | + | − | − |
| Per-Gal4 | + | − | − |
| Lim | + | − | − |
| Cha7.4-Gal4 | − | + | + |
| GH146-Gal4 | − | + | + |
| LN1-Gal4 | − | − | − |
| Acj6 | + | + | + |

Note that the molecular signature of the *otd*⁻/⁻ LALv1 lineage is similar to that of the wild-type ALad1 lineage.

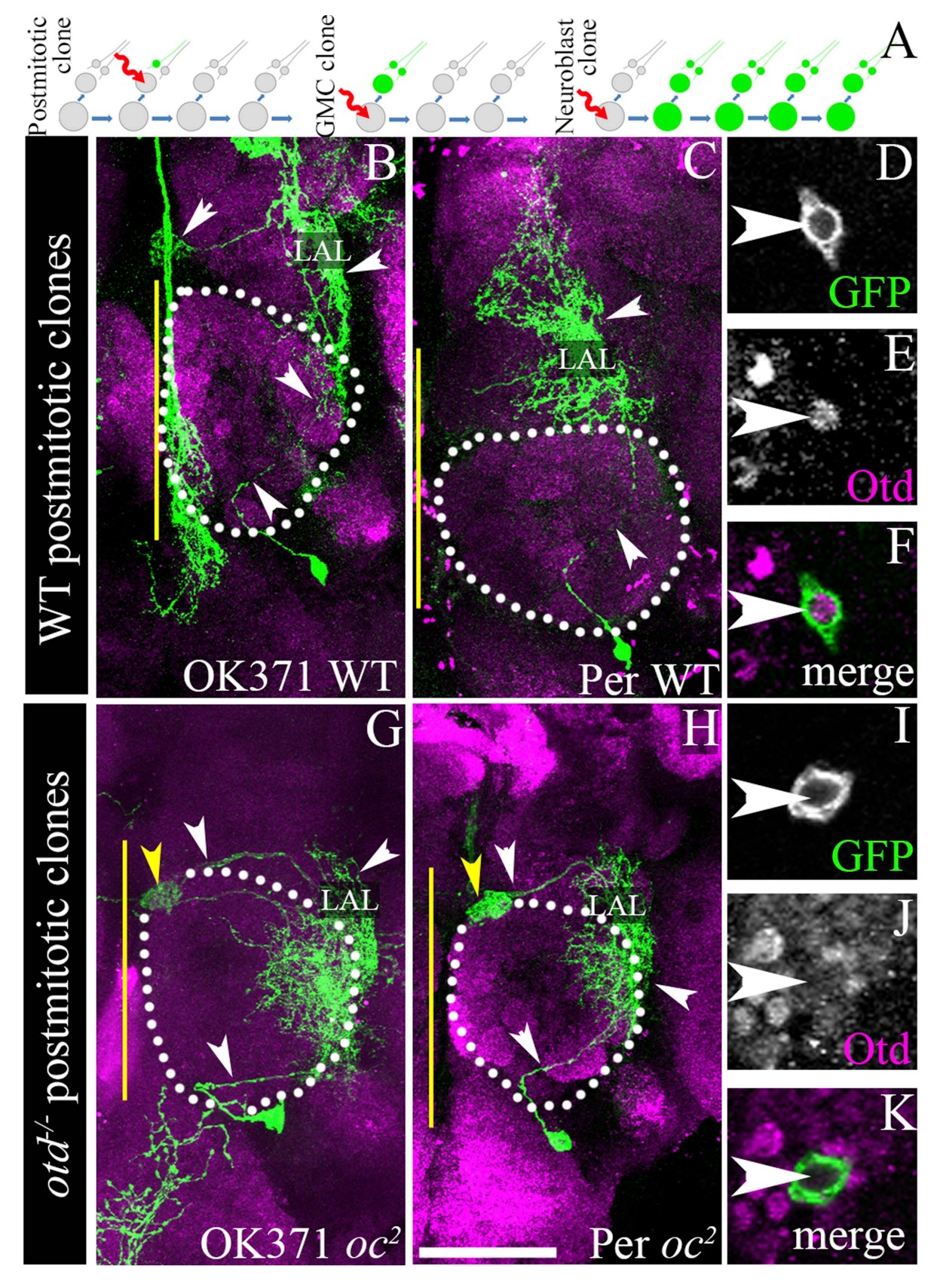

Figure 2. *Continued on next page*

*Figure 2. Continued*

**Figure 2**. *otd$^{-/-}$* postmitotic clones of the LALv1 lineage. (**A**) shows a schematic of clone generation by the MARCM method. (**B** and **C**) document two wild-type single cell MARCM clones of the LALv1 lineage. These neurons skirt around the antennal lobe (white dotted lines) and innervate the LAL. (**D**–**F**) show that these wild-type single cell clones express Otd. (**G** and **H**) document two single cell *otd$^{-/-}$* MARCM clones of the LALv1 lineage. (**I**–**K**) shows one such cell (**I**), which does not express Otd (**J**). Note that the *otd$^{-/-}$* single cell clones of the LALv1 lineage shown in **G** and **H** skirt around the antennal lobe and innervate the LAL and one of the central complex nodule (yellow arrowhead in **G** and **H**), like the wild-type single cell clones shown in **B** and **C**. Genotypes in **B**–**F**: *FRT19A/FRT19A,Tub-Gal80,hsFLP; Per-Gal4 or OK371-Gal4,UAS-mCD8::GFP/+*. Genotypes in **G**–**K**: *FRT19A,oc$^2$/FRT19A,Tub-Gal80,hsFLP; Per-Gal4 or OK371-Gal4,UAS-mCD8::GFP/+*. Scale bar is 50 μm. Yellow line represents the midline.

*otd$^{-/-}$* LALv1 neuroblast clones had dramatic changes in its neuroanatomy (***Figure 4—figure supplement 1B–D***). Mutant neurons no longer innervated the central complex or lateral accessory lobe neuropiles; instead they innervated the antennal lobe neuropile (asterisk in ***Figure 4—figure supplement 1B–D***) and sent projections via the medial antennal lobe tract towards the protocerebrum (arrowhead in ***Figure 4—figure supplement 1B–D***). These changes in dendritic and axonal innervation patterns were reversed by targeted expression of the full-length *otd* coding sequence in mutant neuroblast clones using the *Tub-Gal4* (***Figure 4—figure supplement 2***).

Taken together, these findings indicate that the mutant LALv1 neurons have acquired a transformed identity. Moreover these data suggest that this transformed identity has features characteristic of antennal lobe projection neurons.

## Determining the lineage identity of the transformed neurons

As the neuroanatomy of the *otd$^{-/-}$* LALv1 lineage shows such a dramatic transformation, we wanted to confirm that the transformed neurons did indeed belong to the LALv1 lineage. We used three different approaches to determine that it was indeed the LALv1 lineage that transformed into an antennal lobe-like lineage upon the loss of *otd* from its neuroblast. First, we showed that the appearance of the transformed *otd$^{-/-}$* LALv1 lineage corresponds to the appearance of an extra antennal lobe lineage. Second, we showed that the transformed *otd$^{-/-}$* LALv1 lineage results in the corresponding loss of the LEp tract specific to the wild-type LALv1 lineage. Third, we used an independent molecular marker to unambiguously identify the wild-type and *otd$^{-/-}$* LALv1 lineage.

### An extra antennal lobe lineage

In the wild-type adult brain, the *GH146-Gal4* driver specifically labels a subset of the antennal lobe projection interneurons that derive from three identified neuroblast lineages, ALad1, ALl1, and ALv1 (***Stocker et al., 1997***; ***Jefferis et al., 2001***). If the transformed identity of the *otd$^{-/-}$* LALv1 lineage does indeed correspond to that of an antennal lobe lineage, it should also express the antennal lobe specific enhancer Gal4 driver line, *GH146*. To investigate this, we generated wild-type and *otd$^{-/-}$* MARCM clones in the LALv1 lineage and used the *GH146-Gal4* enhancer line to label the recovered clones. As expected, the only wild-type neuroblast clones recovered corresponded to the ALad1, ALl1, and ALv1 lineages; *GH146-Gal4* labelled wild-type neuroblast clones of the LALv1 lineage were never recovered in these experiments. However, when *otd$^{-/-}$* neuroblast clones were induced in the LALv1 lineage, we recovered 39 examples of a fourth type of *GH146-Gal4*-labelled neuroblast clone. As the expression of *GH146-Gal4* is much more restricted that *Tubulin-Gal4*, this also provided us with the opportunity to describe the *otd$^{-/-}$* LALv1 lineage in more detail. The neurons in this type of mutant clone all exhibited extensive dendritic innervation of the antennal lobe (39/39; ***Figure 4D–G***). This innervation was largely multiglomerular (***Figure 4D–G***), although not all glomeruli were always innervated (yellow arrowhead in ***Figure 4E***) and innervation tended to be more intense in the posterior parts of antennal lobe (***Figure 4G***). Furthermore, the axons of this clone projected via the medial antennal lobe tract towards the protocerebrum (37/39; white arrow in ***Figure 4B,G,H,K***), where they innervated the calyx of the mushroom bodies (39/39; ***Figure 4I,L***) and then turned laterally to innervate the lateral horn (39/39; ***Figure 4J,M***).

The cell body position of this type of *otd$^{-/-}$* clone as well as the entry point of its tract into the antennal lobe (both ventro-medial) did not correspond to any of the known *GH146-Gal4* labelled antennal lobe lineages (***Ito et al., 2013***; ***Yu et al., 2013***). Importantly, this is also true for the ALv1 lineage, whose cell bodies are also located ventral to the antennal lobe; despite the ventral cell body position of the LALv1 and ALv1 lineages, their overall neuroanatomy is very different from each other.

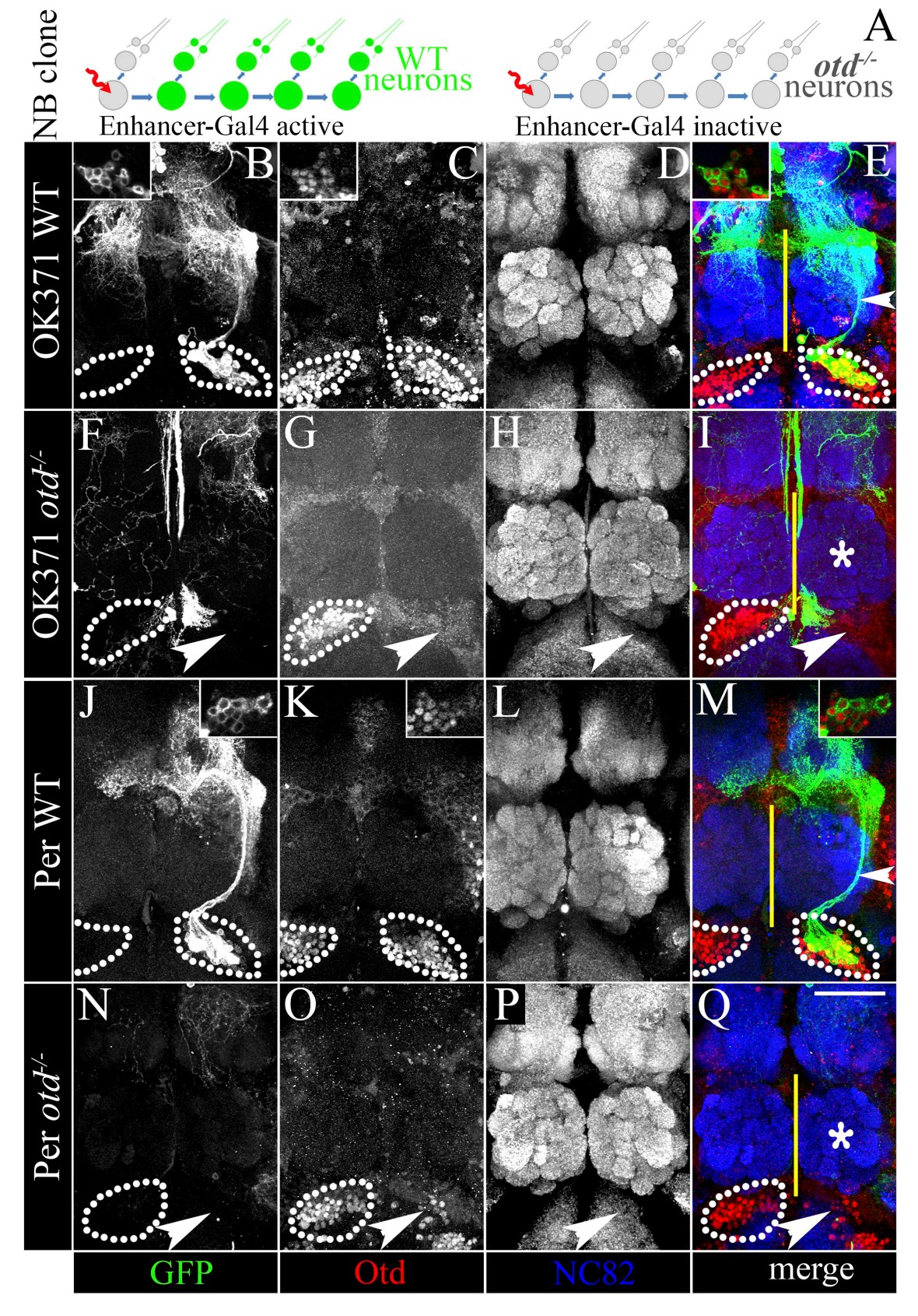

*Figure 3. Continued*

**Figure 3**. Loss of *otd* from the LALv1 lineage results in the suppression of the OK371 and Per enhancers. (**A**) schematises the experimental logic. In neuroblast clones of the LALv1 lineage-specific enhancer Gal4s label the wild-type LALv1 neurons because they are active in the lineage (for example, OK371 and Per). Inability to label the LALv1 neurons in *otd*⁻/⁻ neuroblast clones of the LALv1 lineage (detectable by absence of Otd immunolabelling) will suggest that the enhancers become suppressed in the mutant neurons. The brains in **B–E** and **J–M** show wild-type MARCM clones of the LALv1 lineage labelled by *OK371-Gal4* and *Per-Gal4*, respectively. The insets in **B–E** and **J–M** show that these cells express Otd. The brains in **F–I** and **N–Q** show *otd*⁻/⁻ clones of the LALv1 lineage. Otd expression is lost in one hemisphere in these brains (white arrowhead in **G** and **O**; compare with the Otd expression within white dotted lines in the other brain hemisphere). Neither the *OK371-Gal4* (**F**) nor the *Per-Gal4* (**N**) enhancers drive the expression of UAS-*mCD8::GFP* in these cells and the transformed lineage is not labelled in these experiments. Genotype in **B–E**: *FRT19A/FRT19A,Tub-Gal80,hsFLP; OK371-Gal4,UAS-mCD8::GFP/+*. Genotype in **F–I**: *FRT19A, otd^{YH13}/FRT19A,Tub-Gal80,hsFLP; OK371-Gal4,UAS-mCD8::GFP/+*. Genotype in **J–M**: *FRT19A/FRT19A,Tub-Gal80,hsFLP; Per-Gal4,UAS-mCD8::GFP/+*. Genotype in **N–Q**: *FRT19A, otd^{YH13}/FRT19A,Tub-Gal80,hsFLP; Per-Gal4,UAS-mCD8::GFP/+*. Midline is represented by a yellow line. Scale bar is 50 μm.

The neurites of the *otd*⁻/⁻ LALv1 lineage enter the lobe medially (magenta asterisk in *Figure 5A,D,H*) while the neurites of the ALv1 (as well as the *otd*⁻/⁻ ALv1) enter it laterally (yellow asterisk in *Figure 5A,D,H*). Moreover, while the *otd*⁻/⁻ LALv1 lineage uses the medial antennal lobe tract (magenta arrow in *Figure 5A,D,H*), the ALv1 (as well as the *otd*⁻/⁻ ALv1) uses the mediolateral antennal lobe tract (yellow arrow in *Figure 5A,C*). Finally, while the *otd*⁻/⁻ LALv1 lineage first innervates the calyx of the mushroom body and then the lateral horn, the GH146-labelled neurons of ALv1 (as well as the *otd*⁻/⁻ ALv1) largely innervate only the lateral horn. The *otd*⁻/⁻ LALv1 lineage was also recovered along with each of the three antennal lobe lineages, and in one case all three known *GH146-Gal4* labelled antennal lobe lineages (ALad1, ALl1 and ALv1) were recovered along with it, resulting in four distinct *GH146-Gal4* labelled lineages in the brain (*Figure 5A–C*).

We further confirmed this using the *GH146-QF* driver, which, like the *GH146-Gal4* driver labels the ALad1, ALl1, and the ALv1 lineages (*Potter et al., 2010*). In this background, we used the pan-neuroblast-specific *Insc-Gal4* driver to down regulate Otd expression early in all neuroblasts. As expected, in control brains, the *GH146-QF* driver labelled a total of three antennal lobe lineages ALad1, ALl1, and ALv1 (magenta dotted lines in *Figure 5D–G*). In contrast, in brains where *otd* was efficiently down regulated in all neuroblasts, the *GH146-QF* driver labelled an additional fourth projection interneuron lineage in addition to the three clusters normally seen (*Figure 5H–K*). This confirms that the transformed *otd*⁻/⁻ LALv1 lineage is distinct from the other antennal lobe lineages and results in the addition of an extra projection interneuron lineage in the antennal lobe. Taken together, these data suggest that the loss of *otd* from the LALv1 neuroblast results in the addition of an extra antennal lobe lineage.

## Loss of the LEp tract

If the loss of *otd* from the neuroblast of the LALv1 lineage does indeed result in its neuroanatomical transformation into a lineage of a different fate, then this should correspond to the loss of the LALv1-specific axon tract (LEp) in the brain. To investigate this, we first characterized the axon tract of the wild-type LALv1 lineage, which is readily identifiable in the adult brain based on Neuroglian immunolabelling patterns (*Pereanu et al., 2010*). In wild-type brains, Neuroglian immunolabelling shows the loVM tract (cyan arrow, left hemisphere in *Figure 6C,D*) and the characteristic LEp tract of the LALv1 lineage, which projects around the antennal lobes and towards the central complex (cyan arrowhead, left hemisphere in *Figure 6C,D*). In brains in which one LALv1 neuroblast clone is mutant, the brain hemisphere that contains the *otd*⁻/⁻ LALv1 neuroblast clone (identified by the loss of Otd immunolabelling; yellow dotted lines in *Figure 6B'*), still shows the loVM tract (cyan arrow, right hemisphere in *Figure 6C,D*), which is shared by other lineages. However, the LALv1-specific LEp tract, which projects towards the central complex is entirely missing (cyan arrowhead, right hemisphere in *Figure 6C,D*). This shows that loss of *otd* from the LALv1 neuroblast corresponds to the loss of the LEp tract of the wild-type LALv1 lineage, providing support that loss of *otd* from the LALv1 neuroblast transforms it into a lineage of different identity.

## An independent molecular marker

To investigate the identity of the mutant LALv1 lineage further, we identified Acj6 as a molecular marker that could unambiguously identify the LALv1 lineage in wild-type and *otd* mutants. Acj6 is a

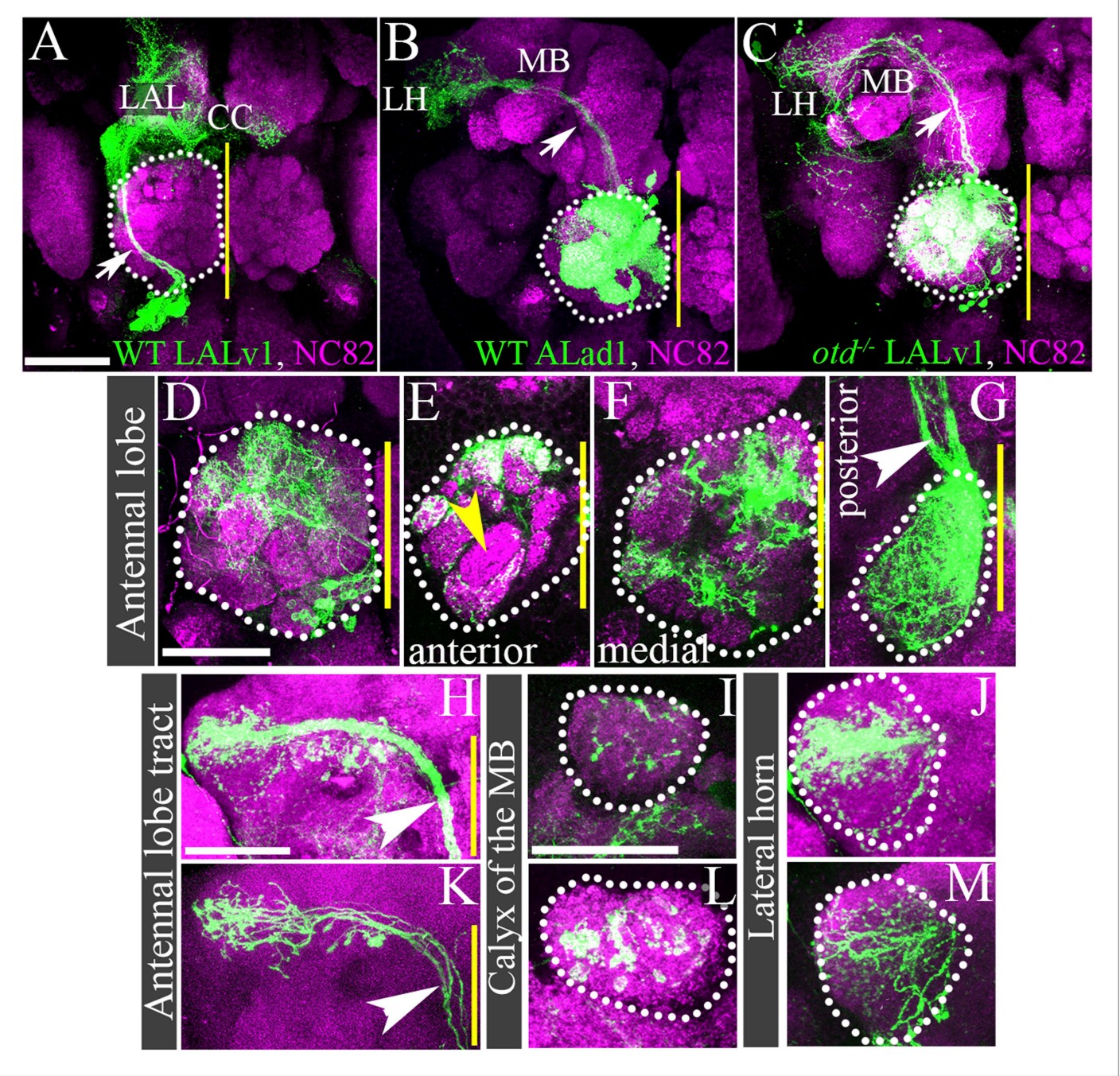

**Figure 4**. Loss of *otd* from LALv1 lineage results in activation of the antennal lobe-specific enhancer, GH146-Gal4, thus allowing detailed study of the neuroanatomy of the transformed *otd⁻/⁻* LALv1 lineage. (**A** and **B**) show WT LALv1 and ALad1 lineages, respectively, and (**C**) shows an *otd⁻/⁻* LALv1 lineage. Note that the neuroanatomy of the *otd⁻/⁻* LALv1 lineage (**C**) is completely unlike the WT LALv1 lineage (**A**) and is strikingly similar to the WT ALad1 lineage (**B**). (**D–M**) document different brains that contain the *otd⁻/⁻* LALv1 lineage. They have multiglomerular dendritic innervations in the AL (**D–G**), but not all glomeruli are always innervated (yellow arrowhead in **E**). There are more innervations in the more posterior parts of the antennal lobe (**F** and **G**). Their axon tracts project via the medial antennal lobe tract (arrowhead in **C, G, H, K**) to innervate the calyx of the mushroom body (MB; **I** and **L**) and the lateral horn (LH; **J** and **M**). Genotype in **A**: *FRT19A/FRT19A,Tub-Gal80,hsFLP; Per-Gal4,UAS-mCD8::GFP/+*. Genotype in **B**: *FRT19A/ FRT19A,Tub-Gal80,hsFLP; GH146-Gal4,UAS-mCD8::GFP/+*. Genotype in **C–M**: *FRT19A,oc² or FRT19A,otd^YH13/FRT19A,Tub-Gal80,hsFLP; GH146-Gal4,UAS-mCD8::GFP/+*. Scale bars are 50 μm (scale bar in **A** is applicable to **B** and **C**; the one in **D** is applicable through to **G**; the one in **H** is applicable to **K**; the one in **I** is applicable to **L, J, M**). Yellow line represents the midline.

*Figure 4. Continued on next page*

*Figure 4. Continued*

The following figure supplements are available for figure 4:

**Figure supplement 1**. Loss of *otd* from the LALv1 neuroblast lineage, transforms it into an antennal lobe PN lineage.

**Figure supplement 2**. Expressing full-length *otd* in the *otd*⁻/⁻ LALv1 lineage rescues the transformation phenotype.

POU transcription factor that is known to be expressed in the ALad1 lineage and in a subset of the ALl1-derived projection interneurons of the wild-type brain (*Figure 7B*). In addition to these two cell clusters, we observed a third Acj6 positive cell cluster ventral to the antennal lobe of the wild-type brain (cyan arrowhead in *Figure 7B*). MARCM clonal labelling using the *Per-Gal4* enhancer line together with anti-Acj6 and anti-Otd antibodies unambiguously identified this cluster as the LALv1 lineage (cyan arrowhead in *Figure 7A–D*). Importantly, this cell cluster continues to express Acj6 immunoreactivity following mutational inactivation of *otd* in the LALv1 lineage (*Figure 7F,G*). Thus, Acj6 provides a molecular marker for the identification of the LALv1 lineage independent of Otd expression in wild-type and mutant clones. The analysis of *otd*⁻/⁻ LALv1 MARCM clones identified by Acj6 immunolabelling and co-labelled by *GH146-Gal4* shows that the fourth *GH146*-positive neuroblast clone described above does indeed correspond to the mutant LALv1 lineage (*Figure 7E,H*). This confirms that upon the loss of *otd* from the neuroblast, the neural progeny of the LALv1 lineage transform into an antennal lobe fate.

Interestingly, the cell body position of the transformed *otd*⁻/⁻ LALv1 neurons varied somewhat in the 76 *otd*⁻/⁻ neuroblast clones obtained (from both the *Tub-Gal4* and *GH146-Gal4* experiments). In most cases (54/76) the cell body position of the *otd*⁻/⁻ LALv1 neuroblast clones remained ventral to the antennal lobe, similar to the cell body position of the wild-type central complex lineage. Thus, in most cases, in terms of cell body position the neuroanatomy did not transform towards the antennal lobe lineage position (antero-dorsal to the adult antennal lobe). Occasionally, however, the cell bodies were shifted closer to the midline (17/76), and in a few rare cases the *otd*⁻/⁻ LALv1 neuronal cell bodies were located antero-dorsal to the antennal lobe (5/76), a position similar to the wild-type antennal lobe lineage (see *Video 3*). This suggests that loss of *otd* in LALv1 neurons not only consistently transform their axonal and dendritic terminals towards the antennal lobe lineage neuroanatomy but also relocate their cell bodies in some cases to resemble the ALad1 antennal lobe projection neuron lineage.

In summary, loss of *otd* from the LALv1 neuroblast results in a transformation of its progeny neurons, from a wild-type central complex identity to an antennal lobe projection neuron identity.

## Overexpression of Otd in the antennal lobe lineage results in a partial reciprocal transformation

We next asked whether *otd* gain-of-function in the antennal lobe lineage, ALad1, might result in a reciprocal anatomical transformation of this lineage into one resembling the wild-type central complex lineage, LALv1. We used the MARCM system to misexpress the full-length *otd* coding sequence in the antennal lobe neuroblast clones using a *Tub-Gal4* driver. In all Otd misexpression clones of the antennal lobe lineage (15/15), we found a partial transformation of this lineage towards the central complex identity (*Figure 8*). All 15 clones comprised a few cells that retained neuroanatomical features of the wild-type antennal lobe lineage such as antero-dorsal cell body position, innervation of the antennal lobe, and axonal projections via the medial antennal lobe tract (yellow asterisk and arrowhead in *Figure 8A–C*). However, most of the cells in the clones displayed neuroanatomical features of the central complex lineage. These cell bodies were positioned ventral to the adult antennal lobe, they projected their axons via the loVM and LEp tracts and they innervated the lateral accessory lobe (magenta asterisk and white arrowheads in *Figure 8A–C*). Thus, *otd* gain-of-function was able to cause a partial, albeit incomplete transformation, of the antennal lobe lineage into a central complex-like lineage.

## Specific molecular changes occur in the transformed *otd*⁻/⁻ LALv1 lineage

The innervation pattern of the *otd*⁻/⁻ LALv1 neuroblast lineage strikingly resembled that of the antennal lobe lineage, ALad1. Furthermore, the projection neuron-specific *GH146* driver, which does

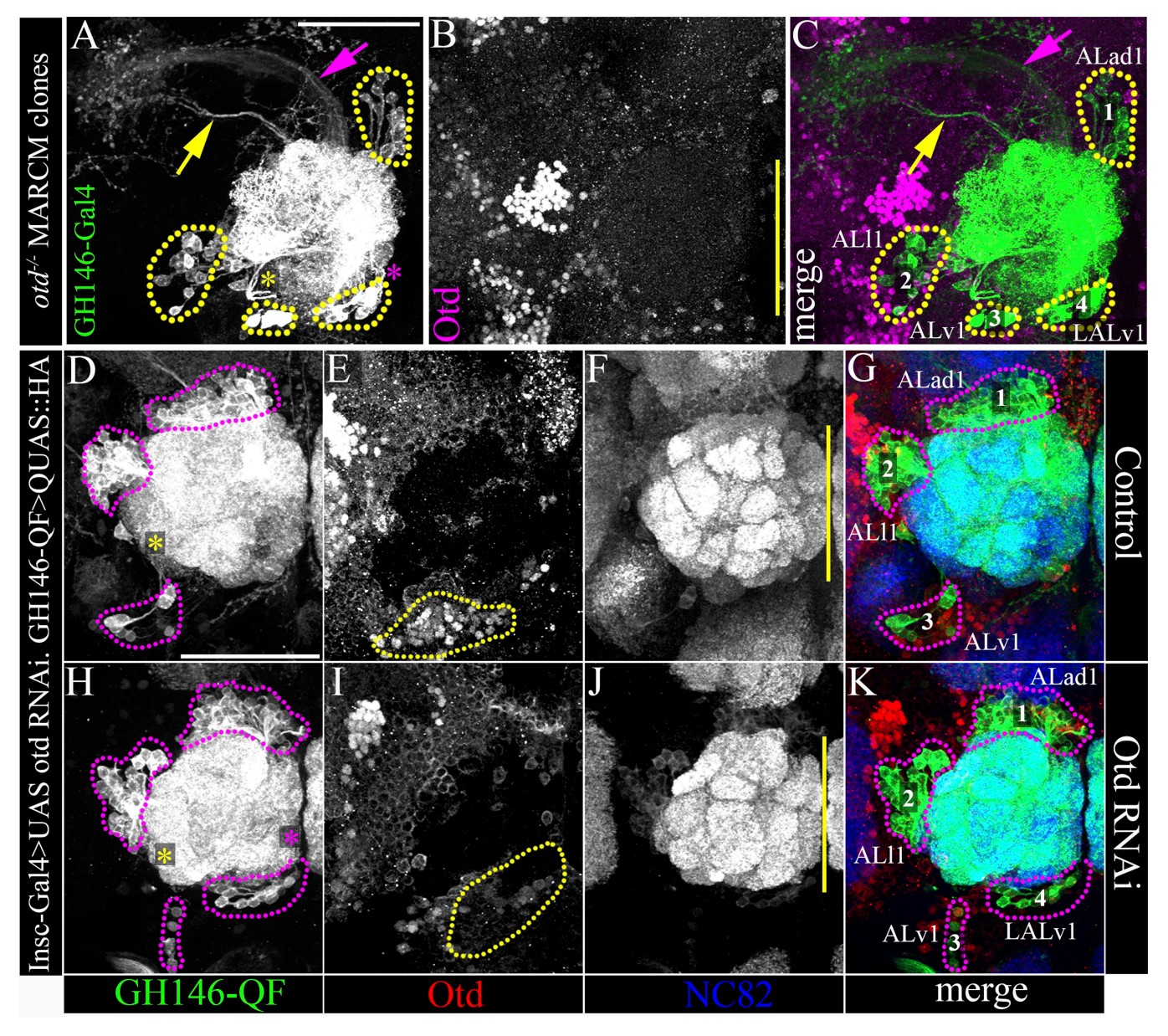

**Figure 5**. Loss of *otd* from the LALv1 lineage results in an extra, fourth antennal lobe lineage labelled by the *GH146* enhancer. (**A–C**) document a MARCM clonal brain, in which the clones null for *otd* function and are labelled by the *GH146-Gal4*. In this brain, the three known antennal lobe lineages normally labelled by *GH146-Gal4* (ALad1, ALl1, and Alv1) have been recovered, along with an additional fourth neuroblast lineage, LALv1. (**D–G**) document a control brain where Otd is not down regulated in the LALv1 lineage (yellow dotted lines in **E**). In this brain, there are three clusters of antennal lobe projection neurons labelled by *GH146-QF* corresponding to the ALad1, ALl1 and ALv1 lineages, magenta dotted lines in **D** and **G**). (**H–K**) show a brain in which there has been efficient knock down of Otd in the LALv1 lineage (note loss of Otd immunolabelling ventral to the antennal lobe; yellow dotted lines in **I**). In this brain, apart from the ALad1, ALl1, and ALv1 projection neurons, an additional, fourth cluster of cells is seen innervating the antennal lobe ('4', LALv1). The yellow asterisks in **A**, **D**, **H** indicate the point of entry of the ALv1 lineage into the antennal lobe, and the yellow arrow indicates its axon tract. The magenta asterisks in **A**, **D**, **H** indicate the point of entry of the LALv1 lineage into the antennal lobe, and the magenta arrow indicates its axon tract. Note that these are distinct from each other. Genotype in **A–C**: *FRT19A,otd^{YH13}/ FRT19A,Tub-Gal80,hsFLP; GH146-Gal4,UAS-mCD8::GFP/+*. Genotype in **D–K**: *UASdicer/+;Insc-Gal4/UAS-miRNA-otd-1;GH146-QF,QUAS-mtdTomato-HA/+*). Scale bars are 50 μm. The one in **A** is applicable to **B** and **C** and the one in in **D** is applicable to **D–K**. Yellow line represents the midline.

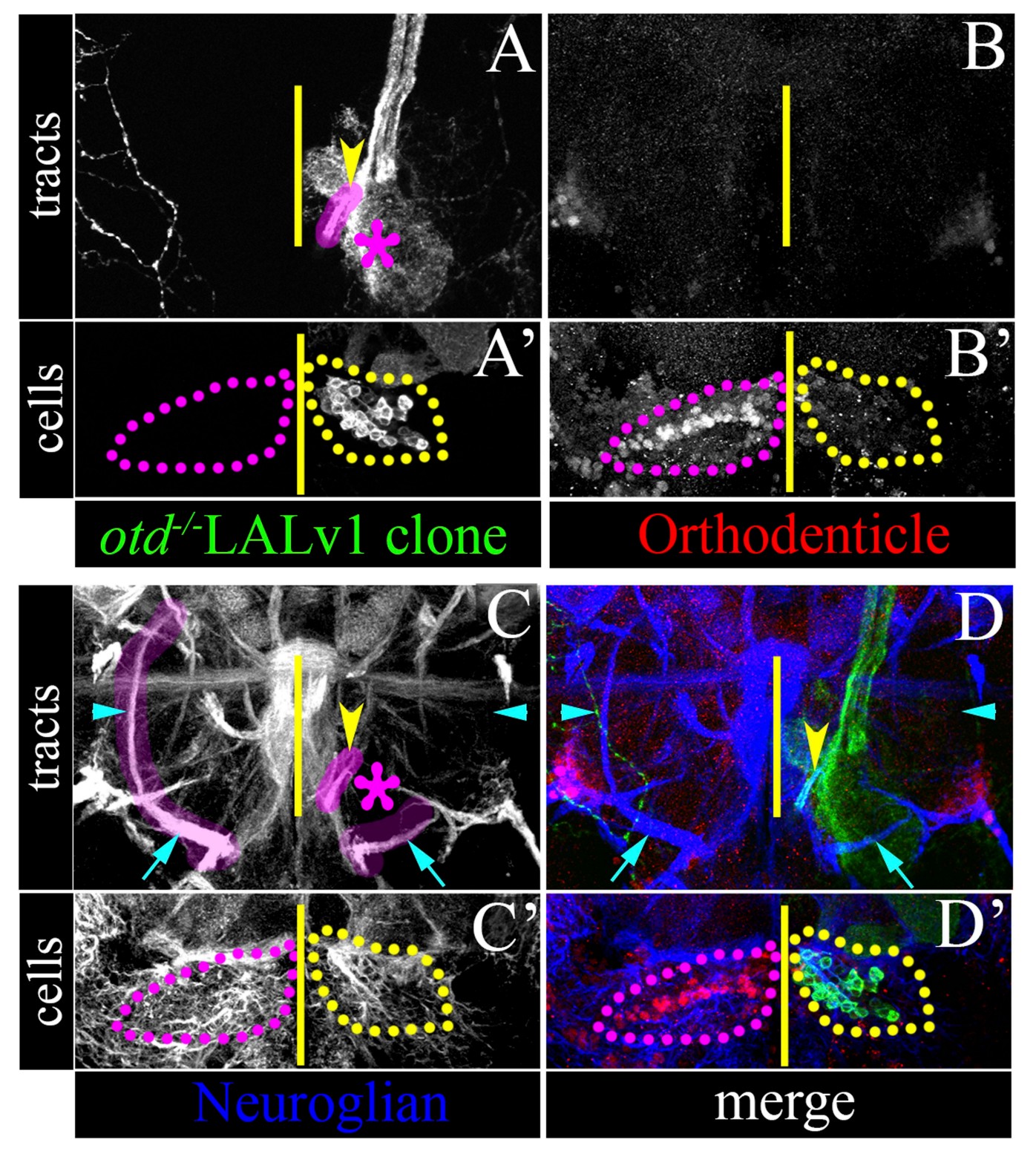

**Figure 6**. Loss of *otd* from the LALv1 lineage results in the absence of the LEp tract. (**A–D**) and (**A'–D'**) document a brain with a single MARCM labelled *otd*⁻/⁻ LALv1 lineage. (**A–D**) show more posterior sections that contain the tracts of the LALv1 lineage; (**A'–D'**) show more anterior sections of the same brain that documents its cell bodies. The right brain hemisphere contains an *otd*⁻/⁻ LALv1 lineage as seen by the loss of Otd immunolabelling ventral to the antennal lobe (yellow dotted lines in **B'**), while the left hemisphere contains a wild-type LALv1 lineage (hence not labelled by MARCM) as seen by the

*Figure 6. Continued on next page*

*Figure 6. Continued*

presence of Otd immunolabelling (magenta dotted lines in **B**'). In the left brain hemisphere, which contains the wild-type LALv1 lineage, the loVM (cyan arrow on the left in **C**) and the LALv1 specific LEp tracts (cyan arrowhead on the left in **C**) that are identifiable by Neuroglian immunolabelling (highlighted in magenta). In the right brain hemisphere, which contains the *otd*⁻/⁻ LALv1 lineage, the loVM tract (taken by other lineages) is still present (cyan arrow on the right in **C**). The LALv1 specific LEp tract (cyan arrowhead on the right in **C**) that is exclusively made by the LALv1 lineage, is entirely missing in the right brain hemisphere, which contains the *otd*⁻/⁻ LALv1 lineage (cyan arrow on the right in **C**). The yellow arrowheads in **A**, **C**, **D** point to the new tract of the *otd*⁻/⁻ LALv1 lineage innervating the antennal lobe (magenta asterisk). Genotype: *FRT19A, otd^YH13/FRT19A,Tub-Gal80,hsFLP; GH146-Gal4,UAS-mCD8::GFP/+*. The midline is represented by a yellow line in all images.

not label the wild-type central complex lineage, labelled the transformed *otd*⁻/⁻ LALv1 neurons. This ectopic activation of the *GH146* driver in the *otd*⁻/⁻ neurons indicates that the anatomical transformation of the mutant neurons might be accompanied by corresponding molecular transformations that reflect a transformation to the antennal lobe lineage identity. To test if this is indeed the case, we analyzed the activity of select molecular markers (the Gal4 driver lines described above) in the central complex lineage (summarized in *Table 1*). In order to test the activity of these enhancers in the *otd*⁻/⁻ LALv1 lineage, we generated *otd*⁻/⁻ MARCM clones and used these selected *Gal4* lines to label the mutant lineage (see schematic in *Figure 3A*). Finally, we also tested the expression of the homeodomain transcription factor LIM1, which is expressed in the wild-type LALv1 lineage and is not expressed in the wild-type ALad1 lineage.

As described above, when we generated wild-type MARCM clones and used either *OK371-Gal4* or *Per-Gal4* driver lines to label the clones, we found that both lines were able to drive reporter expression in the wild-type LALv1 lineage (*Figure 3B–E,J–M*). In contrast, neither of these driver lines was able to drive reporter expression in the *otd*⁻/⁻ LALv1 lineage (*Figure 3F–I,N–Q*). This suggests that the *OK371-Gal4* and *Per-Gal4* driver lines are suppressed in the transformed *otd*⁻/⁻ LALv1 neurons.

However, in these experiments, the transformed neurons were not labelled at all because the activity of the drivers was suppressed in the *otd*⁻/⁻ LALv1 lineage. In order to confirm this finding, we decided to positively label the *otd*⁻/⁻ LALv1 neurons and in this background assay the activity of the *Gal4* drivers. In order to do this, we utilized the dual MARCM method (*Lai and Lee, 2006*), which uses two independent binary expression systems (Gal4-UAS and LexA–LexA operator) to label the MARCM clones. In these experiments, we used the *GH146-LexA* driver to label the *otd*⁻/⁻ LALv1 neurons positively and combined it with *OK371-Gal4* and *Per-Gal4* driver lines to assay their activities. We first tested if the *GH146-LexA* driver, like the *GH146-Gal4* and the *GH146-QF* driver, was active in the *otd*⁻/⁻ LALv1 neurons. Thus, in the first set of dual MARCM experiments, we combined the *GH146-LexA* with *Tubulin-Gal4* (*Figure 9A–D*). Under these conditions, when we generated *otd*⁻/⁻ neuroblast clones in the LALv1 lineage, we found that the transformed neurons were labelled by both the *Tubulin-Gal4* (magenta dotted lines and inset in *Figure 9A*) and *GH146-LexA* (magenta dotted lines and inset in *Figure 9B*) drivers, confirming that *GH146-LexA*, like *GH146-Gal4* and the *GH146-QF*, is active in the *otd*⁻/⁻ LALv1 neurons and thus able to label it. In the following dual MARCM experiments, we used the *GH146-LexA* to positively label the transformed *otd*⁻/⁻ LALv1 neurons and combined it with either *OK371-Gal4* or *Per-Gal4* driver lines to assay for their activity in the transformed neurons. In both cases, while the *GH146-LexA* positively labelled the transformed *otd*⁻/⁻ LALv1 neurons (magenta dotted lines and insets in *Figure 9E,I*), neither of the Gal4 driver lines was able to drive reporter expression in these mutant neurons (magenta dotted lines and insets in *Figure 9F,J*). This indicates that enhancers that are normally active in the wild-type central complex lineage and inactive in the antennal lobe lineage become suppressed in the transformed *otd*⁻/⁻ LALv1 lineage.

Might the converse be true; do enhancers that are normally inactive in the wild-type central complex lineage and active in the antennal lobe lineage become activated in the transformed *otd*⁻/⁻ LALv1 lineage? To test this, we used the *Cha7.4-Gal4* in MARCM clonal experiments. As expected, we never recovered *Cha7.4-Gal4* labelled LALv1 neuroblast clones in wild-type MARCM experiments (data not shown). In contrast, when we generated *otd*⁻/⁻ neuroblast clones in the LALv1 lineage (identifiable by the loss of Otd staining ventral to the antennal lobe; magenta dotted lines in *Figure 9N*) the *Cha7.4-Gal4* driver robustly drove reporter expression in the transformed *otd*⁻/⁻ LALv1 lineage (magenta dotted lines and insets in *Figure 9M–P*). This suggests that the *Cha7.4-Gal4* becomes activated in the transformed *otd*⁻/⁻ LALv1 lineage. Furthermore, the concomitant loss of the *OK371-Gal4* driver (a putative glutamatergic label) and ectopic activation of the *Cha7.4-Gal4* driver (a putative

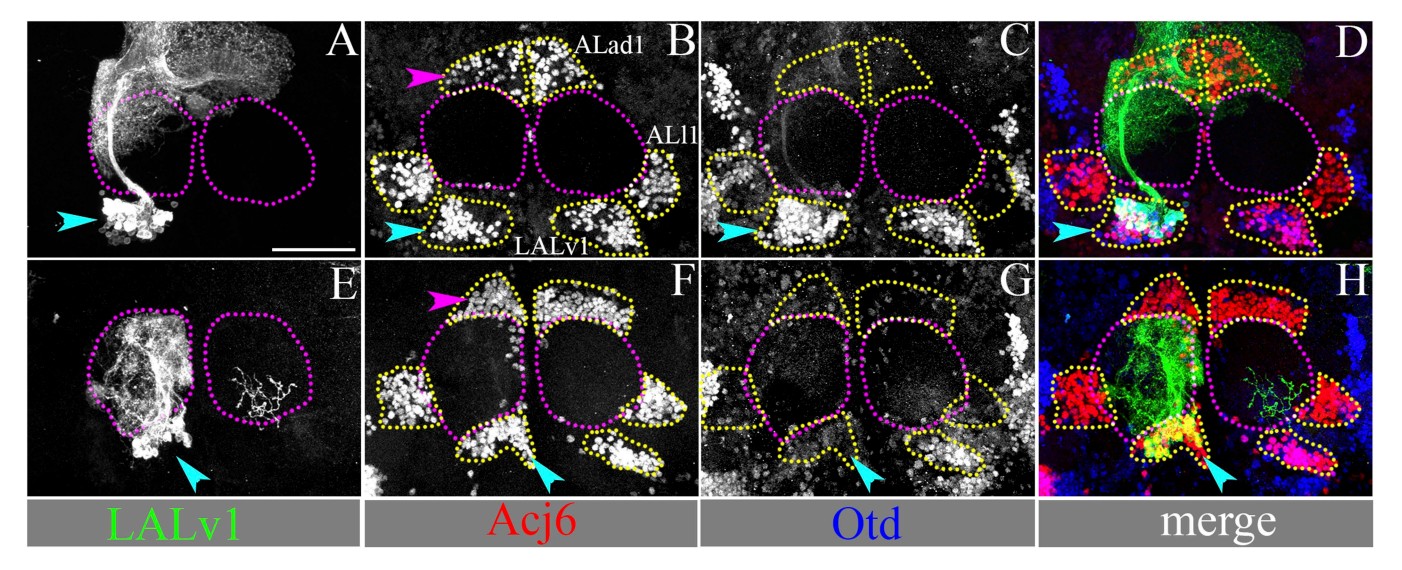

**Figure 7**. The Acj6-positive cell cluster ventral to the AL is the LALv1 lineage and when null for *otd*, innervates the AL. (**A–D**) document a WT MARCM clone of the LALv1 lineage labelled with *Per-Gal4*. This brain has been immunolabelled with Acj6 (**B**) and Otd (**C**). In **B**, only three clusters of Acj6 positive cells (yellow dotted lines) are seen around each of the ALs (yellow dotted lines). The clusters that are ventral to AL (cyan arrowhead) belong to the LALv1 lineage as seen by the WT MARCM clone shown in **A**. As expected, these cells are also Otd positive (cyan arrowhead in **C**). (**E–H**) show an *otd*−/− LALv1 MARCM clone. This brain is also immunolabelled with Acj6 (**F**) and Otd (**G**). Note that the *otd*−/− LALv1 cells, which have a transformed neuroanatomy (cyan arrowhead in **E**), are null for *otd* (cyan arrowhead in **G**) but are identifiable as the ventral cluster of Acj6 positive cells (cyan arrowhead in **F**). Genotype in **A–D**: *FRT19A/FRT19A,Tub-Gal80,hsFLP; Per-Gal4,UAS-mCD8::GFP/+*. Genotype in **E–H**: *FRT19A, otd^{YH13}/FRT19A,Tub-Gal80,hsFLP; GH146-Gal4,UAS-mCD8::GFP/+*. Scale bar is 50 μm.

cholinergic label) suggest that there might be a change in neurotransmitter identity of the LALv1 lineage from wild-type glutamatergic to transformed cholinergic identities, similar to the cholinergic identity of wild-type antennal lobe neurons.

Finally, immunolabelling with an anti-LIM antibody indicates that the wild-type central complex lineage expresses the homeodomain transcription factor LIM (*Figure 10G–I*) while the transformed *otd*−/− LALv1 lineage, like the wild-type antennal lobe lineage, does not (*Figure 10J–L*). Interestingly, the *LN1-Gal4* driver, which is inactive in both the wild-type LALv1 and ALad1 lineages, remains inactive in the *otd*−/− LALv1 lineage (data not shown).

Taken together, these findings indicate that the *otd*−/− LALv1 lineage acquires the molecular signature of a wild-type antennal lobe lineage (see *Table 1*) implying that *otd* loss-of-function in the LALv1 neuroblast lineage results in a molecular as well as an anatomical transformation of this lineage into one resembling the ALad1 lineage.

### The *otd*−/− transformed LALv1 lineage establishes functional connectivity in the antennal lobe and can respond to odour stimulation

Given the extent of the anatomical and molecular transformations seen in the *otd*−/− LALv1 neuroblast lineage, might the neurons in the transformed lineage receive functional input from olfactory sensory neurons? To address this question, we specifically expressed the calcium sensor G-CaMP3 in the transformed *otd*−/− LALv1 lineage by MARCM clonal labeling (*Wang et al., 2003*) and used two-photon microscopy to monitor calcium activity in the dendrites of these transformed neurons in the antennal lobe.

We first tested if the transformed *otd*−/− LALv1 neurons established functional connectivity with other neurons in the antennal lobe. Typically, olfactory sensory neurons bring odour information to the antennal lobe via the antennal nerve. Here, they make synaptic connections with projection neurons and local interneurons; projection neurons, take the odour information to higher brain centres (mushroom body and lateral horn) and local interneurons process the odour information locally in the antennal lobe. We reasoned that if the transformed neurons did made functional connections within

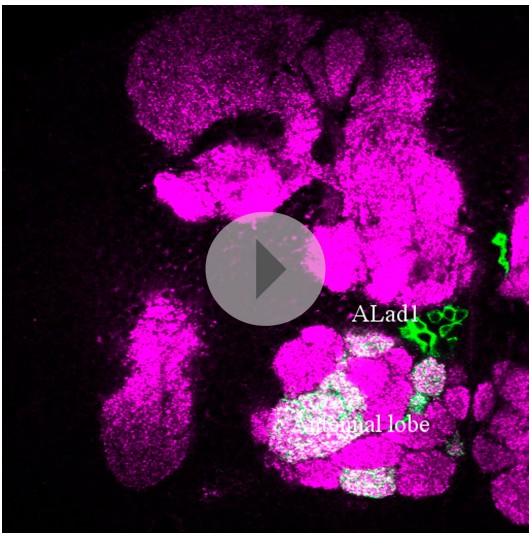

**Video 1**. Projection pattern of the wild-type ALad1 lineage.

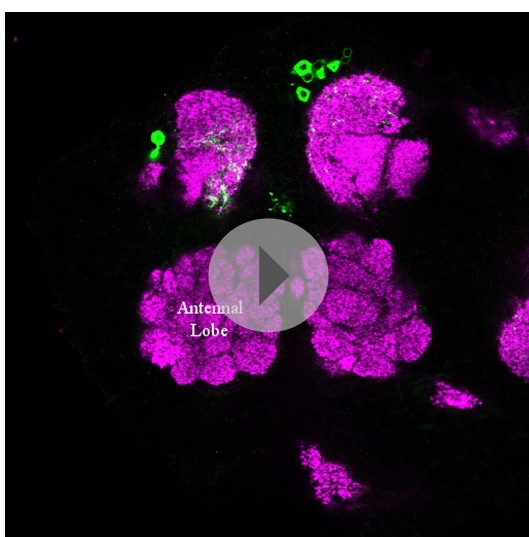

**Video 2**. Projection pattern of the wild-type LAlv1 lineage.

the antennal lobe they would be postsynaptic to the olfactory sensory neurons and would be activated upon the activation of the antennal nerve. We therefore electrically stimulated the antennal nerve while simultaneously monitoring calcium activity from the transformed neurons. We found that electrical stimulation of the antennal nerve, which contains the axons of the olfactory sensory neurons, resulted in an increase in calcium activity in the dendrites of the *otd*⁻/⁻ LALv1 lineage. Moreover, a greater number of electrical stimulus pulses applied to the antennal nerve resulted in a corresponding increase in the amplitude of the calcium signal recorded in the mutant transformed neurons (***Figure 11A–C***). These results demonstrate that the transformed *otd*⁻/⁻ LALv1 neurons were able to make functional connections in the antennal lobe and were able to receive functional input from sensory afferents.

We further investigated if the transformed *otd*⁻/⁻ LALv1 neurons could respond to specific odour stimuli. To do this, we performed calcium imaging experiments similar to those described above but replaced the electrical stimulation of the antennal nerve with odour stimulation of the intact antenna (olfactory sensory neurons). Four different odorants (isoamyl acetate, ethyl butyrate, 3-octonal, 3-heptanol) were selected based on their ability to excite all or some of the VM2, DM2 and DM3 glomeruli (***Dacks et al., 2009***; ***Semmelhack and Wang, 2009***) in the antennal lobe, which are innervated by the *otd*⁻/⁻ LALv1 neurons. Imaging calcium activity from the dendrites of the *otd*⁻/⁻ LALv1 neurons in these glomeruli in response to the selected odours show that each of the four odorants evoked a unique pattern of glomerular activity. Isoamyl acetate excited all three glomeruli, whereas ethyl butyrate excited only the VM2 and DM2 glomeruli. 3-octanol and 3-heptanol, however, excited just the DM2 and VM2 glomeruli, respectively (***Figure 11D,E***). These patterns of glomerular activation in the *otd*⁻/⁻ LALv1 neurons are strikingly similar to that of wild-type antennal lobe olfactory projection neurons (***Wang et al., 2003***; ***Dacks et al., 2009***; ***Semmelhack and Wang, 2009***).

Taken together, these functional studies indicate that the *otd*⁻/⁻ LALv1 neurons receive specific input from olfactory sensory neurons that results in glomerulus-specific activation patterns to different odorants. This in turn implies that *otd* loss-of-function in a single neuroblast leads to a remarkably extensive reconfiguration of the macrocircuitry in the brain, which includes anatomical, molecular as well as functional transformation of neurons in the central complex lineage into neurons with properties of olfactory projection neurons.

## Discussion

During neuronal development in both vertebrates and invertebrates neural progenitors use spatial and temporal information to generate diverse neuronal subtypes. For example, in *Drosophila*, unique spatial information imparts heterogeneity to the neuroblast pool and then temporal cues acting in the

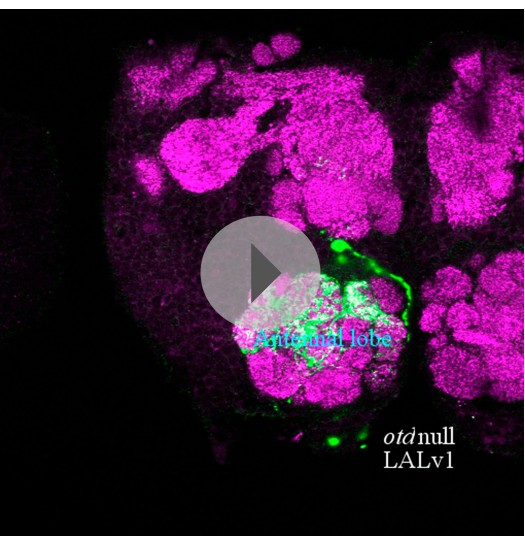

**Video 3**. Projection pattern of the otd⁻/⁻ LAlv1 lineage.

neuroblasts generate further diversity. In this way diverse neuronal subtypes are produced by the neuroblast lineages, which consequently create the diverse functional macrocircuitry of the brain. In addition, the neuroblasts in the central brain of *Drosophila* are characterized by the expression of a specific combination of cell intrinsic determinants (*Urbach and Technau, 2003a*) that are thought to act in the specification of unique neuroblast identity and hence control lineage-specific neuronal cell fate.

In this study, we show that such intrinsic determinants present in the neuroblast are essential for the proper specification of the entire lineage that derives from the neuroblast. Our data demonstrate that a remarkable rewiring of the functional macrocircuitry of the brain occurs due to the manipulation of one intrinsic factor, *otd*, acting in an identified neuroblast during development. This transformation affects molecular properties, anatomical projection patterns (dendritic and axonal), and functional inputs in all of the neurons in the lineage (summarized in *Figure 12*) such that a central complex lineage is transformed into a functional olfactory projection neuron lineage. This *otd*-dependent, lineage-specific respecification of interneurons has implications for our understanding of the development and evolution of the circuitry in the brain.

The ability of a neuroblast lineage to transform completely into another upon the loss of a single intrinsic determinant suggests that many of the other putative members of a potential neural identity

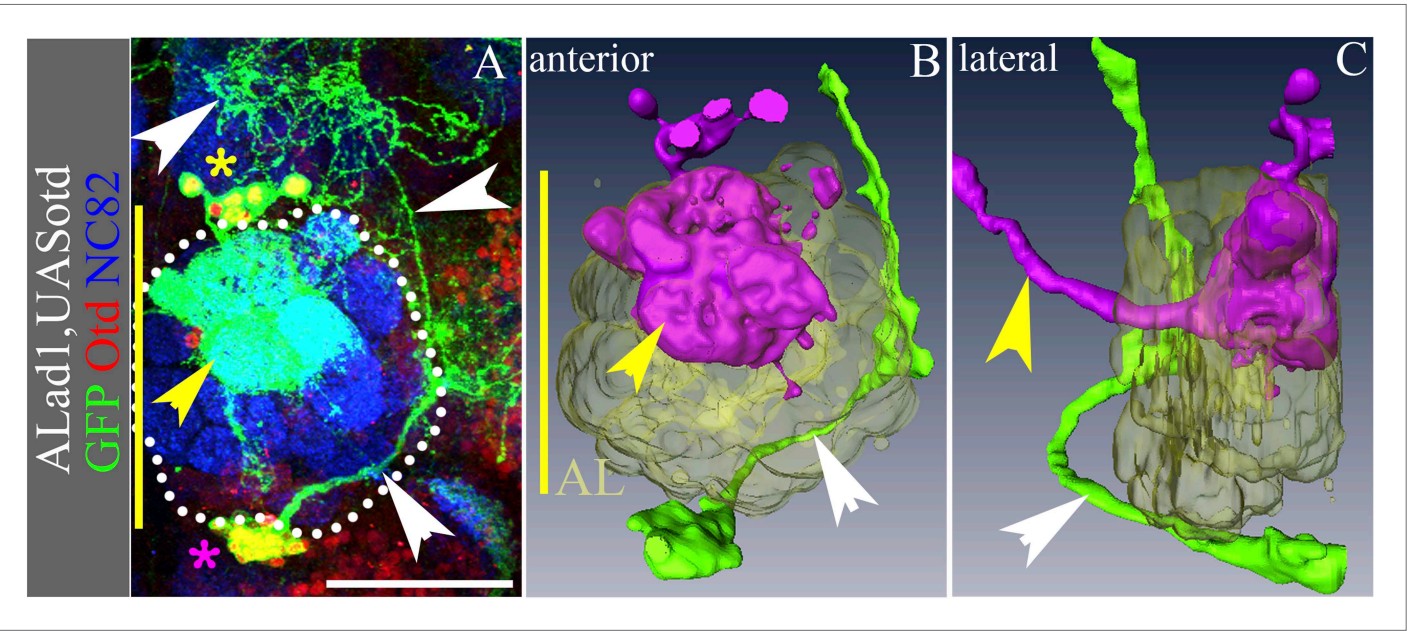

**Figure 8**. Overexpression of Otd in the ALad1 lineage results in a partial reciprocal transformation. (**A–C**) ALad1 neuroblast clone misexpressing Otd. (**B** and **C**) are the anterior and lateral views of the 3D reconstructions of the clone in **A**. Few of the Otd misexpressing cells (yellow asterisk in **A**) retain the wild type neuroanatomy of the ALad1 lineage (magenta cells in the reconstructions)—they have innervations in the AL glomeruli (yellow arrowheads in **A–C**) and project via the antennal-cerebral tract (yellow arrowhead in **C**). Most of the Otd misexpressing ALad1 neurons are seen ventral to the AL (magenta asterisk in **A**; green cells in the reconstructions in **B** and **C**). They do not innervate the AL and instead project towards the LAL (white arrowheads in **A–C**). Genotype in **A–C**: *FRT19A/FRT19A,Tub-Gal80,hsFLP; Tub-Gal4,UAS-mCD8::GFP/UAS-otd*. Midline is represented by a yellow line. Scale bar is 50 μm.

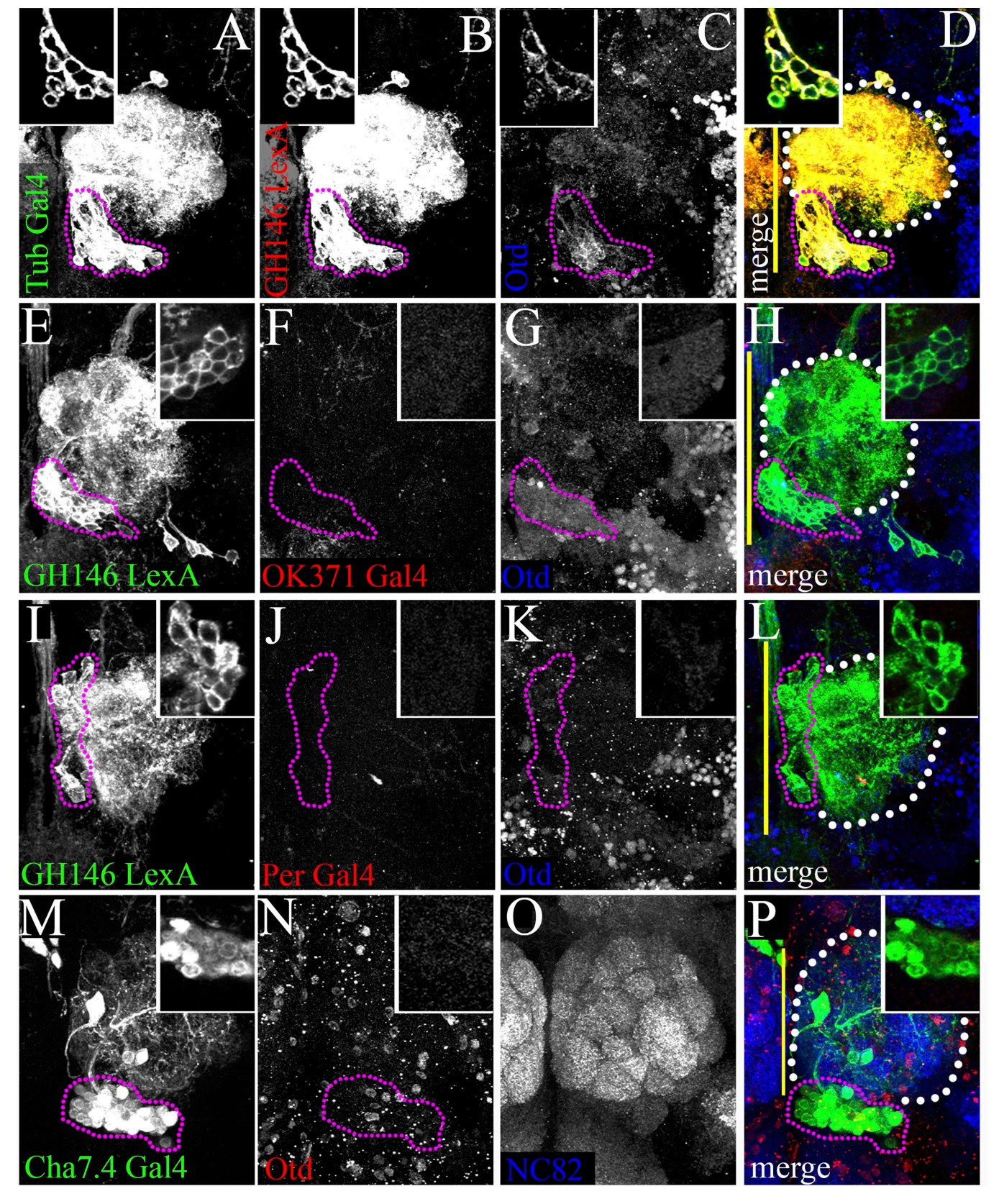

Figure 9. Continued on next page

*Figure 9. Continued*

**Figure 9**. Specific molecular changes occur in the transformed *otd⁻/⁻* LALv1 lineage. (**A–D**) documents an *otd⁻/⁻* LALv1 neuroblast clone that has been dual labelled with *Tubulin-Gal4* (**A**) and *GH146-LexA* (**B**) using the dual MARCM technique. (**E–H** and **I–L**) document *otd⁻/⁻* LALv1 neuroblast clones that have been dual labelled with *GH146-LexA* (**E** and **I**) to positively label the mutant neurons and either *OK371-Gal4* (**E–H**) or *Per-Gal4* (**I–L**) using the dual MARCM technique. In these clones, while the *GH146-LexA* driver labels the transformed *otd⁻/⁻* LALv1 neurons (**E** and **I**), neither the *OK371-Gal4* (**F**) nor the *Per-Gal4* (**J**) do. (**M–P**) documents a transformed *otd⁻/⁻* LALv1 lineage (inset and magenta dotted lines in **N**) labelled with *Cha7.4-Gal4*. While *Cha7.4-Gal4* is not active in the wild-type LALv1 lineage (not shown), it becomes activated in the *otd⁻/⁻* LALv1 lineage (inset and magenta dotted lines in **M–P**). Genotypes in **A–D**: *FRT19A, otd^YH13^/FRT19A,Tub-Gal80,hsFLP; Tubulin-Gal4,UAS-mCD8::GFP/FRTG13,GH146-LexA::GAD,LexAop::rCD2::GFP*. Genotype in **E–H**: *FRT19A,otd^YH13^/FRT19A,Tub-Gal80,hsFLP; OK371-Gal4,UAS-mCD8::GFP/FRTG13,GH146-LexA::GAD,LexAop::rCD2::GFP*. Genotype in **I–L**: *FRT19A,otd^YH13^/FRT19A,Tub-Gal80,hsFLP;Per-Gal4,UAS-mCD8::GFP/FRTG13,GH146-LexA::GAD,LexAop::rCD2::GFP*. Genotype in **H**: *FRT19A,otd^YH13^/FRT19A,Tub-Gal80,hsFLP; Cha7.4-Gal4,UAS-mCD8::GFP/+*. Midline is represented by a yellow line.

code might be shared between these lineages. The observation that the neuroblasts of the central complex and antennal lobe lineages develop in such close spatial proximity to each other during early development suggests that these two neuroblasts may experience similar spatial cues as they develop on the neuroectoderm. If this is the case, then by manipulating a single differentially expressed factor, *otd*, we might have been able to uncover the underlying similarity in the intrinsic spatial code between the two neuroblast lineages. Importantly, this neuroblast-specific transformation of lineage identity resulted in an alteration of the brain's circuitry such that an entire neuroanatomical unit of projection to the central complex was lacking while a novel and functional ectopic unit of projection was added to the antennal lobe.

Implicit in these findings are the notions of 'coded' and 'soft' properties of circuit assembly. On the one hand, the neuroanatomical and molecular transformation described above demonstrates that circuitry in the brain is 'hard-wired' or 'coded' by the spatially encoded intrinsic factors—the presence or absence of *otd* from the central complex neuroblast determines the identity of the resultant neurons. On the other hand, the resulting functional transformation suggests that circuit assembly involves substantial 'soft-wiring'—the olfactory sensory neurons and interneurons indigenous to the antennal lobe are able to make functional connections with the extraneous transformed *otd⁻/⁻* LALv1 neurons, which they are normally not 'hard-wired' to connect with. Thus, while genetically encoded properties might 'lock' lineages into particular circuit states (central complex or antennal lobe) it is their 'soft' properties (developmental plasticity) that allow circuits to functionally incorporate changes as dramatic as extraneous neurons. As both these wiring strategies operate simultaneously in the brain, it bestows upon the brain the huge potential of evolvability of functional circuits.

Many interesting questions emerge as a result of our findings. How does the developmental plasticity of a functional circuit support these large-scale rearrangements? Do developing circuits acquire a propensity for exuberant connectivities, or do they try and maintain a homeostasis in their connections and therefore make compensatory changes in the number of synapses with their normal partners? It has been shown in some cases that neuronal activity can mediate such 'soft' properties of synaptic connections (*Tripodi et al., 2008*; *Singh et al., 2010*). It will be interesting to test if this is also the case for the transformed neurons and the olfactory circuit. Finally, do all parts of the brain display such striking developmental plasticity such that they can be remodelled to this extent and incorporate extraneous neurons into existing circuitry?

The ability to change the functional macrocircuitry of the brain through changes in the expression of a single transcription factor in a single neuroblast lineage may provide a simple paradigm for large-scale modification of brain connectivity during evolution. The *otd⁻/⁻*transformed LALv1 lineage functionally integrates into the antennal lobe circuitry and participates in olfactory information processing. This suggests that a functional rewiring of the olfactory circuitry can occur due to the addition of an entire neuroblast lineage to the normal olfactory circuit. In more general terms, this type of lineage-specific rewiring might fuel the evolutionary modification of neural circuitry in the brain. It provides an elegant and simple solution to the evolution of complex circuitry in that a 'microevolutionary' molecular change (changing the expression of one gene in one cell) can have 'macroevolutionary' consequences on the brain's circuitry (changing an entire macrocircuit or an information processing module). This simple strategy suggests that large-scale changes in the brain's wiring do not need to come about through many minor, sequentially accumulating changes at the cellular level. Instead, large-scale wiring changes can occur in response to remarkably simple changes in gene expression in single cells. In this

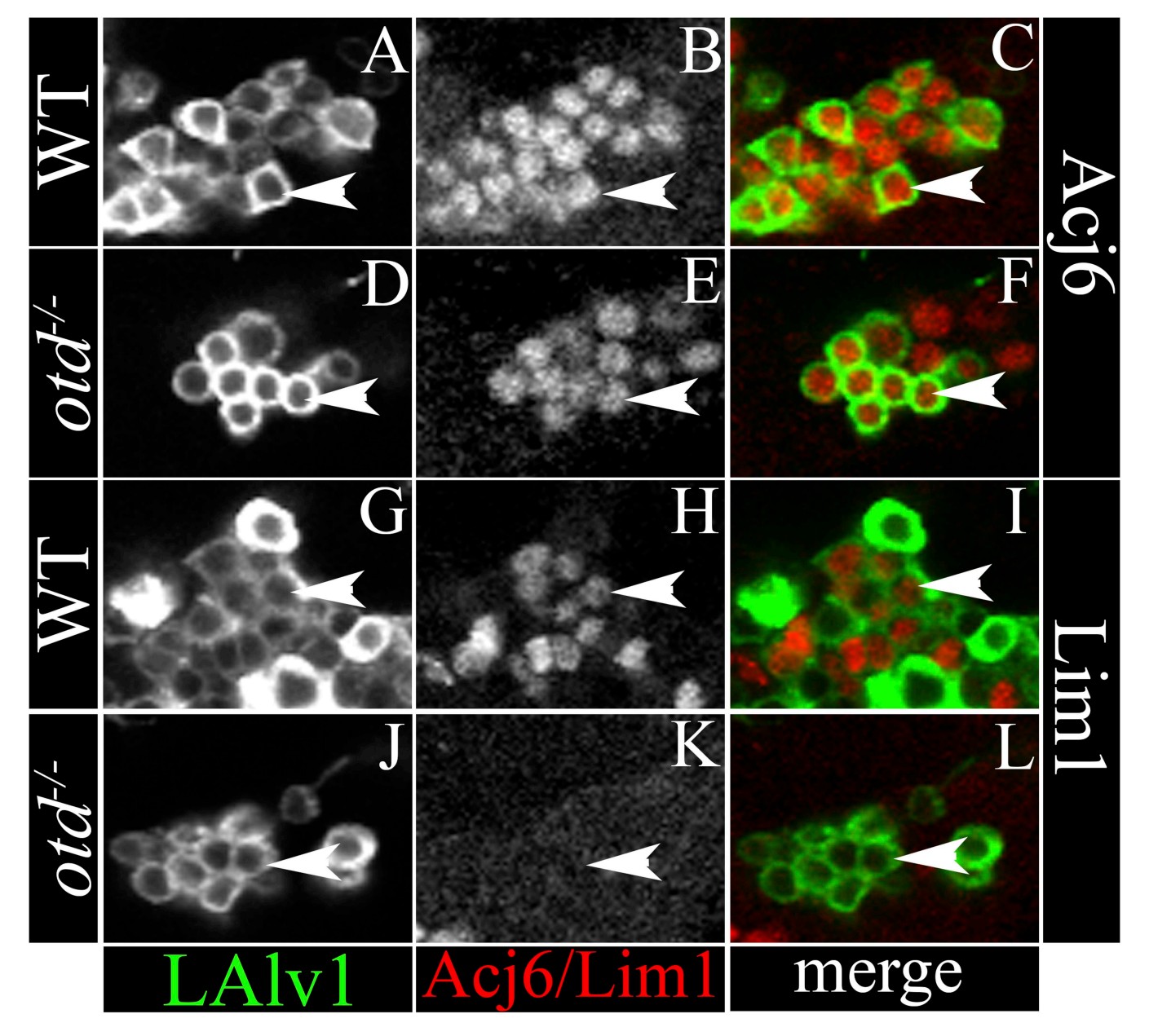

**Figure 10**. Specific molecular changes occur in the transformed *otd*⁻/⁻ LALv1 lineage. (**A–C** and **G–I**) document neuronal cell bodies of the wild-type LALv1 lineage immunolabelled with Acj6 (**A–C**) and Lim1 (**G–I**). The WT LALv1 neurons express both these molecular markers. (**D–F** and **J–L**) document neuronal cell bodies of the *otd*⁻/⁻ LALv1 lineage immunolabelled with Acj6 (**D–F**) and Lim1 (**J–L**). While *otd*⁻/⁻ LALv1 neurons continue to express Acj6 (**E**), they downregulate Lim1 expression (**K**).

context, our data may have provided evidence for the mechanistic ease with which circuitry in the brain can evolve.

## Materials and methods

### Lineage nomenclature

The brain lineages have been named by various groups in the past. Here, we list the various names by which the two lineages, which are the focus of our study, have been called. ALad1 (*Ito et al., 2013*; *Yu et al., 2013*)/BAmv3 (*Pereanu and Hartenstein, 2006*; *Lovick et al., 2013*; *Wong et al., 2013*)/adNB

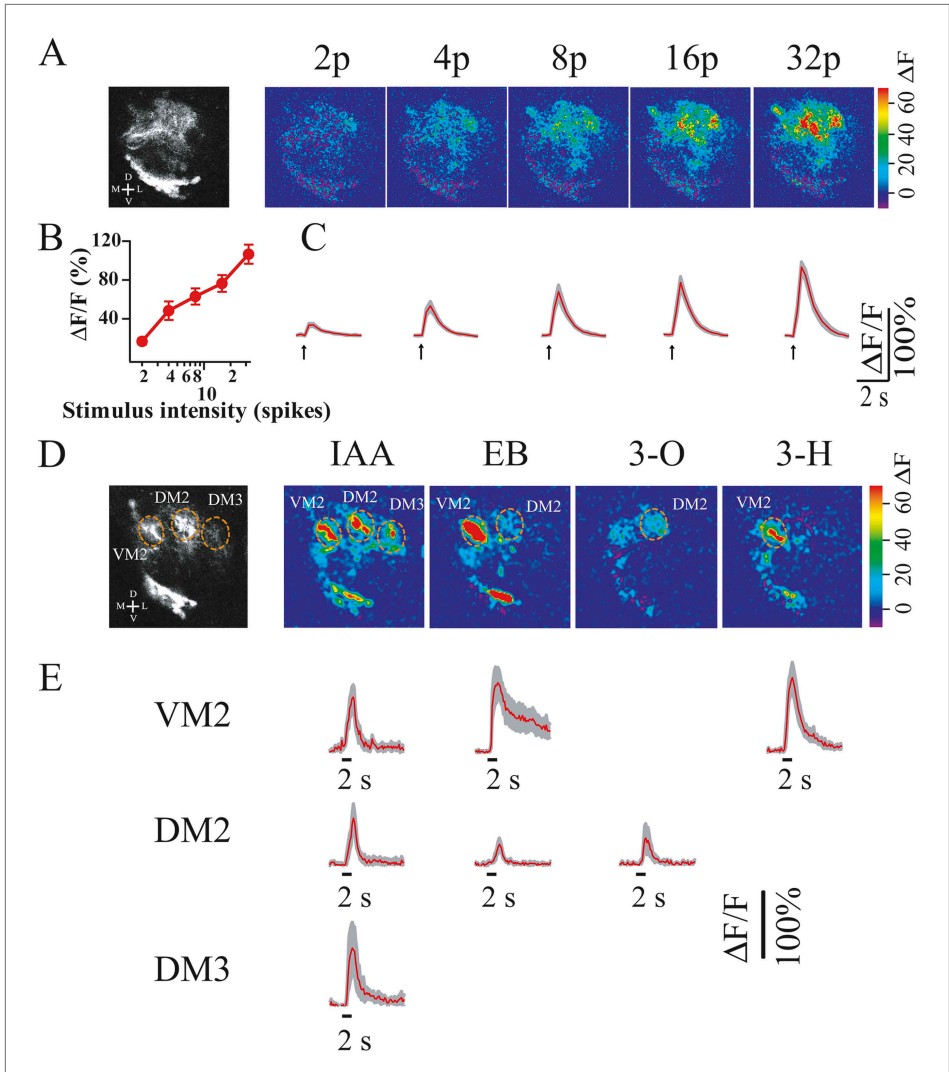

**Figure 11**. The *otd*[−/−] transformed LALv1 lineage has functional synapses in the AL and can respond to odour stimulation. (**A**) Two-photon calcium imaging from the *otd*[−/−] LALv1 neurons in response to electrical stimulation of the antennal nerve. Gray-scale image shows average pre-stimulation fluorescence of one transformed lineage. Colour images show fluorescence changes in response to different numbers of electrical stimulation (duration, 1 ms; amplitude, 10 V; frequency, 100 Hz). Graph in **B** shows peak amplitude of ΔF/F against the number of stimulations. Error bars represent S.E.M. n = 5. (**C**) Average traces that plot ΔF/F over time at different stimulus intensities. Shaded region in each trace represents S.E.M. (**D**) Two-photon calcium imaging from the *otd*[−/−] LALv1 neurons in response to odour stimulation of the antenna. Gray-scale image shows the structure of the transformed lineage and three identifiable glomeruli. The colour images show glomerular activation patterns evoked by isoamyl acetate (IAA), ethyl butyrate (EB), 3-octanol (3-O) and 3-heptanol (3-H). IAA activated VM2, DM2 and DM3. EB activated both VM2 and DM2. 3-O and 3-H activated DM2 and VM2, respectively. (**E**) Average traces that plot ΔF/F over time in VM2, DM2 and DM3. The shaded region in each trace represents S.E.M. n = 3–5. The false colour scales (ΔF) and scale bars are shown at the right of each panel. Genotype: *FRT19A, otd[YH13]/FRT19A,Tub-Gal80,hsFLP;GH146-Gal4, UAS-GCaMP3/GH146-Gal4,UAS-GCaMP3*.

---

(*Jefferis et al., 2001*). LALv1 (*Ito et al., 2013*; *Yu et al., 2013*)/BAmv1 (*Pereanu and Hartenstein, 2006*; *Lovick et al., 2013*; *Wong et al., 2013*).

## Fly strains and genetics

Fly stocks were obtained from the Bloomington Stock Centre (IN, USA) and, unless otherwise stated, were grown on cornmeal media, at 25°C. UAS-miRNA otd-1 was kindly provided by Henry Sun, Taiwan.

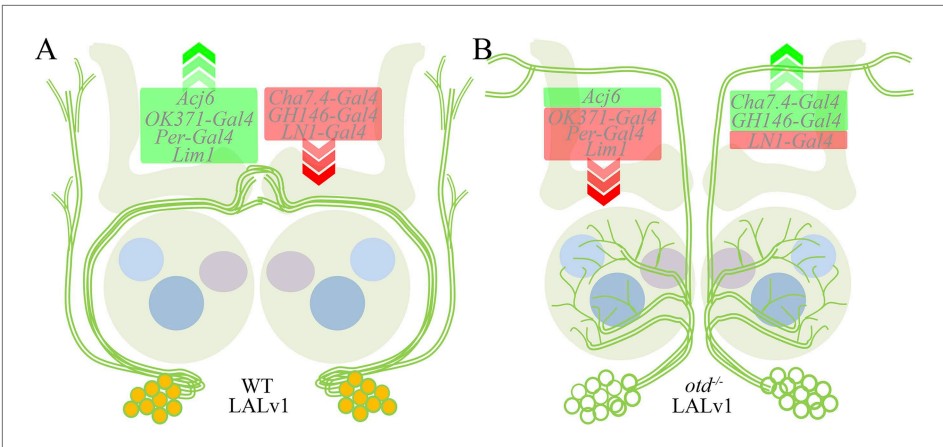

**Figure 12**. Transformation of the LALv1 lineage. (**A**) The WT LALv1 is a wide-field central complex lineage that expresses Otd (orange). (**B**) The loss of *otd* function from the LALv1 neuroblast transforms these neurons into antennal lobe projection neurons. This neuroanatomical transformation is accompanied by molecular changes—markers that are active in the WT LALv1 (green box in **A**) are suppressed in the mutant (**B**; with the exception of Acj6); those that are suppressed in the WT (red box in **A**), are activated in the mutant (**B**; with the exception of LN1-Gal4). The exceptions are in agreement with a transformation to antennal lobe projection neuron fate (see *Table 1*).

To generate WT MARCM clones females of the genotype *FRT19A* or *FRT19A;UAS-mCD8::GFP* were crossed to males of the following genotypes: *FR19A,Tub-Gal80,hsFLP; Tub-Gal4,UAS-mCD8::GFP/CyO* or *FR19A,Tub-Gal80,hsFLP; OK371-Gal4,UAS-mCD8::GFP/CyO* or *FR19A,Tub-Gal80,hsFLP; Per-Gal4,UAS-mCD8::GFP/CyO* or *FR19A,Tub-Gal80,hsFLP; GH146-Gal4,UAS-mCD8::GFP/CyO* or *FR19A,Tub-Gal80,hsFLP; LN1-Gal4,UAS-mCD8::GFP/CyO* or *FR19A,Tub-Gal80,hsFLP; Cha7.4-Gal4,UAS-GFP/CyO.* To generate *otd*$^{−/−}$ MARCM clones, females of the genotype *FRT19A,otd*$^{YH13}$/*FM7c* or *FRT19A,oc*$^2$/*FM7c* or *FRT19A, otd*$^{YH13}$;*UAS-mCD8::GFP* or *FRT19A,oc*$^2$;*UAS-mCD8::GFP* were crossed to males of the above-mentioned genotypes.

To knockdown *otd* in neuroblasts using RNAi and simultaneously visualise GH146-labelled cells, females of the genotype *UAS-dicer; Insc-Gal4/CyO* were crossed to males of the genotype *UAS-miRNA otd-1/CyO; GH146-QF,QUAS-mtdTomato-HA/TM6B* and the crosses were grown at 25°C.

To generate Dual MARCM clones, females of the genotype *FRT19A,otd*$^{YH13}$/*FM7a; FRTG13, GH146-LexA::GAD,LexAop-rCD2::GFP/CyO* were crossed to males of the following genotypes: *FRT19A,Tub-Gal80,hsFLP; Tub-Gal4,UAS-mCD8::GFP/CyO* or *FRT19A,Tub-Gal80,hsFLP; OK371-Gal4, UAS-mCD8::GFP/CyO* or *FRT19A,Tub-Gal80,hsFLP; Per-Gal4,UAS-mCD8::GFP/CyO.*

To generate Otd overexpression clones, females of the genotype *FRT19A;UAS-otd* were crossed to males of the genotype *FR19A,Tub-Gal80,hsFLP; Tub-Gal4,UAS-mCD8::GFP/CyO.* To generate rescue clones, females of the genotype *FRT19A otd*$^{YH13}$/*FM7a;UAS-otd* were crossed to males of the genotype *FR19A,Tub-Gal80,hsFLP; Tub-Gal4,UAS-mCD8::GFP/CyO.*

To generate clones for functional imaging, females of the genotype *FRT19A,otd*$^{YH13}$/*FM7a;GH146-Gal4,UAS-GCaMP3/CyO* were crossed to males of the genotype *FR19A,Tub-Gal80,hsFLP; GH146-Gal4,UAS-GCaMP3/CyO.*

Embryos were collected from the progeny of all the clonal crosses and these were given a heat shock for 1 hr at 37°C at either the embryonic stage, 0–4 hr after larval hatching or 24 hr after larval hatching.

To knock down Otd in neuroblasts using RNA interference, females of the genotype *UASdicer; Insc-Gal4* were crossed to males of the genotype *UAS-miRNA-otd-1;GH146-QF,QUAS-mtdTomato-HA/TM6B* and grown at 25°C.

## Generation of UAS-Otd

A full-length *otd*-cDNA for expression under the control of the UAS promoter was made in the KVR lab by cloning a full-length *otd* cDNA including the 5′ and 3′ UTRs into the pJRFC-MUH vector. The

full-length o*td* cDNA clone RE-10280 (pBluscript backbone) was purchased from DGRC. Confirmed plasmid DNA (pJFRC-10xUAS-Otd FL with an attB integration site) was microinjected (2 µg/µl in 1× microinjection buffer) into *Drosophila* embryos that contain PhiC31 integrase and selected attP docking site on the second chromosome. Further crossing of G0 flies, screening of the transformants and balancing of insertions performed at the Transgenic Fly Facility at C-CAMP facility at NCBS campus, Bangalore, India.

## Immunohistochemistry and imaging

Brains were dissected in 1× PBS and fixed in freshly prepared 4% PFA for 30 min at room temperature. The fixative was removed, and the brains were washed with blocking solution (1× PBS with 0.3% TritonX and 0.1% BSA). Primary antibodies were diluted in blocking solution. The samples were incubated in a moist chamber on horizontal shaker at 4°C for 24 hr. Samples were then washed with 0.3% PTX (1× PBS with 0.3% TritonX) and secondary antibody diluted in 0.3% PTX was added. The samples were incubated in this at 4°C in a moist chamber on horizontal shaker overnight, after which they were washed and mounted in vectashield on glass slides.

Primary antibodies used were: rabbit anti-GFP (1:10,000; Molecular Probes, Invitrogen, Delhi, India); chick anti-GFP (1:10,000, AbCam, Cambridge, UK); mouse anti-Bruchpilot (mAbnc82, 1:20; DSHB, Iowa, USA); Rb anti-Otd (1:1,500, gift from Henry Sun, IMB, Sinica, Taiwan); guinea pig anti-Otd (1:7,500, gift from Tiffany Cook, Cincinnati, OH); mouse anti-Acj6 (1:25, DSHB, Iowa, USA); rabbit anti-dLim1 (1:1000; a gift from J Botas); rat anti-mCD8 (1:100, Caltag, Burlingname, Californina); mouse anti-rCD2 (AbD Serotec, Raleigh, NC, USA); mouse anti-Neurotactin (BP106, 1:10, DSHB, Iowa, USA); mouse anti-Neuroglian (BP102, 1:10, DSHB, Iowa, USA); rabbit anti-HA (1:100, AbCam). Secondary antibodies used were Alexa Fluor-488-, Alexa Fluor-568- and Alexa Fluor-647-coupled antibodies generated in goat (1:400; Molecular Probes, Invitrogen, Delhi, India) and TO-PRO -3 iodide-661 (Thermo Fischer Scientific, MA,USA).

All samples were imaged on Olympus Fluoview (FV1000) laser scanning confocal microscope. Optical sections were acquired at 1-µm intervals with a picture size of 512 × 512 pixels. Images were digitally processed using Adobe Photoshop CS3. 3-D reconstructions were made using Amira.

## Functional imaging

Brains of clonal animals were dissected in $Ca^{2+}$-free AHL saline, which contains 108 mM NaCl, 5 mM KCl, 4 mM $NaHCO_3$, 1 mM $NaH_2PO_4$, 8.2 mM $MgCl_2$, 5 mM HEPES, 5 mM trehalose, and 10 mM sucrose, with pH adjusted to 7.4. Live brains that contain only the *otd* null transformed neurons were selected for imaging experiments.

### Two-photon calcium imaging and antennal nerve stimulation (electrical)

Two-photon calcium imaging was performed as described previously (*Root et al., 2008*). The antennal nerves were cut from the base of the antennae. The brain preparation was then pinned down on a Sylgard-coated petri dish with AHL saline containing 2 mM $CaCl_2$. The antennal nerve was stimulated electrically with a glass suction electrode, at 1 ms in duration, 10 V in amplitude and 100 Hz in frequency (S48 Grass stimulator). The response of the transformed neurons was monitored by a custom-built two-photon microscope. Excitation wavelength was 930 nm. Images were captured at 4 frames/s.

### Two-photon calcium imaging and odour stimulation

For odour stimulation, the fly brain was dissected leaving the antennae intact and embedded in agarose containing AHL saline with 2 mM $CaCl_2$. The agarose gel was removed from the antennae and a piece of Kimwipes was used to dry the antennae. A glass coverslip was placed on top of the brain preparation for imaging. Odour delivery was controlled by solenoid valves described previously (*Root et al., 2008*). Odour vapour was obtained by placing a filter paper containing 10 µl of an odorant in a 100-ml bottle. Mixing an air stream with an odour stream at different flow ratios was used to deliver odorants at a specific concentration. Isoamyl acetate and ethyl butyrate were delivered at 1% (odorant at 10 ml/min and air at 990 ml/min), whereas 3-heptanol and 3-octanol were delivered at 2.5% saturated vapor pressure. Each odorant was applied for a duration of 2 s. Images were acquired for 20 s at a rate of 4 frames/s and a resolution of 128 × 128 pixels. At the end of each experiment, an image stack was collected at a resolution of 512 × 512 pixels for glomerulus identification. The data were analyzed and plotted using Igor Pro 6.2 (Wavemetrics). The peak response of stimulation (ΔF/F) was shown as mean ± S.E.M.

## Acknowledgements

We dedicate this work to the memory of our late friend and colleague Veronica Rodrigues (1953–2010). We acknowledge DST, Govt. of India–Centre for Nanotechnology (no. SR/S5/NM-36/2005) for CIFF at NCBS. We are grateful to Henry Sun and Tiffany Cook for their generous gift of reagents.

## Additional information

### Competing interests

KVR: Senior editor, *eLife*. The other authors declare that no competing interests exist.

### Funding

| Funder | Grant reference number | Author |
| --- | --- | --- |
| Tata Insitute of Fundamental research | National Centre for Biological Sciences | K VijayRaghavan |
| École Polytechnique Fédérale de Lausanne | Indo-Swiss Bilateral Research Initiative | Heinrich Reichert, K VijayRaghavan |
| Swiss National Science Foundation | 31003A 140607 | Heinrich Reichert |
| National Institutes of Health | DK092640 | Jing W Wang |
| Company of Biologists | | Sonia Sen |
| Bristol-Myers Squibb | | Sonia Sen |
| Department of Science and Technology, India | JC Bose Fellowship | K VijayRaghavan |

The funders had no role in study design, data collection and interpretation, or the decision to submit the work for publication.

### Author contributions

SS, DC, RC, SB, Conception and design, Acquisition of data, Analysis and interpretation of data, Drafting or revising the article; JWW, HR, KVR, Conception and design, Analysis and interpretation of data, Drafting or revising the article

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
