## [Decision Letter]

Thank you for sending your work entitled “Rewiring the brain: Genetic transformation of structural and functional circuitry in the *Drosophila* brain” for consideration at *eLife*. Your article has been favorably evaluated by Fiona Watt (Senior editor), a Reviewing editor, and 2 reviewers.

This paper provides a comprehensive analysis of a genetic manipulation, consisting in removing or ectopically expressing the gene orthodenticle (otd), which resulted in the fate transformation of a neural lineage in the Drosophila brain. The authors show convincingly that manipulating otd at the stage of the stem cell (neuroblast) determines the fate of the lineage; loss of otd in a lineage that normally innervates the central complex (LALv1) results in the transformation of its neurons into a lineage that normally develops at a close by position, associated with the antennal lobe). The data are clearly presented and documented. The findings are of great significance for the field. They underline the early global function of transcription factors in fate specification. They also shed light on the interplay between intrinsic (“hard wired”) factors and plasticity; for example, otd-mutant axons adopting the antennal projection fate are able to branch and properly innervate the antennal lobe, even though they start out at a different location (the location of the LALv1 lineage).

Substantial concerns:

1) The interpretations depend on the ability to recognize the same lineage after it has been profoundly transformed. It was difficult for me to be assured of this because of how the paper was organized. The study involves the use of a number of different driver lines that are employed in various contexts throughout the paper. In most cases we are not told why a given driver line was selected for a particular experiment or, importantly, the normal expression pattern for each line with respect to LALv1 and ALad1 and other lineages with which they might be confused. For example, for the RNAi experiments, why select insc-GAL4? What is the selectivity of the line? Also, the Per-Gal4 line was used when examining the single neuron otd-/- clones of LALv1 but GH146-Gal4 was used for the neuroblast clones of the lineage. It is only when we get to Figure 6 do we see that two drivers that express in wild-type LALv1 cells (Per-Gal4 and OK371-Gal4) no longer express in these neurons after they are transformed by otd removal. This information needs to be brought up earlier in the paper.

2) Comparison of 4B and F shows a single cluster of otd+ cells that coincide with LALv1 in “B”, but in “F” part of the weakened otd+ cluster is now ascribed to two different clusters: part to LALv1 [yellow outline] and the remainder to ALv1 [which should not express otd?]. I am not convinced that we are seeing 2 distinct clusters rather than a split version of ALv1. It would be good to have a more convincing example of the appearance of a distinct GH146 cluster with the reduction in otd.

3) The ALv1 lineage is a potential complication because it is a ventral lineage of antennal lobe projection neurons that is situated next to LALv1 in the adult. These cells have an anatomy that is similar to the cells of the transformed LALv1 and the GH146 driver expresses in these cells. The authors should have recovered many ALv1 clones in their experiments. It is important to know how they distinguished ALv1 clones from the transformed LALv1.

4) There are other pieces of information that would help establish the correspondence of the LALv1 cells to the LALv1otd- neurons: 1) The observation that the LEp tract is missing when the lineage is transformed is a compelling piece of information. In addition to the reconstruction [or instead of it], it would be good to show an appropriate confocal section showing the loss of the tract on the transformed side and its presence on the control side. 2) If they have them, it would be good to show images of the transformed clones in the larval stage. The position of the neuroblast and its cell cluster should be in its normal position but its axon bundle would then be taking the transformed trajectory.

---

## [Author Response]

*1) The interpretations depend on the ability to recognize the same lineage after it has been profoundly transformed. It was difficult for me to be assured of this because of how the paper was organized. The study involves the use of a number of different driver lines that are employed in various contexts throughout the paper. In most cases we are not told why a given driver line was selected for a particular experiment or, importantly, the normal expression pattern for each line with respect to LALv1 and ALad1 and other lineages with which they might be confused. For example, for the RNAi experiments, why select insc-GAL4? What is the selectivity of the line? Also, the Per-Gal4 line was used when examining the single neuron otd-/- clones of LALv1 but GH146-Gal4 was used for the neuroblast clones of the lineage. It is only when we get to*
Figure 6
*do we see that two drivers that express in wild-type LALv1 cells (Per-Gal4 and OK371-Gal4) no longer express in these neurons after they are transformed by otd removal. This information needs to be brought up earlier in the paper*.

We thank the reviewers for bringing this to our attention. We have now reorganized the manuscript so as to first mention the different Gal4 drivers used (see paragraph 4 under the sub heading, ‘Development, morphogenesis and differential expression of Otd in two identified central brain neuroblast lineages, LALv1 and ALad1’). We have also taken care to mention the Gal4 drivers used for each of the experiments when we describe them. For example, see second sentence in paragraph 2 and third sentence in paragraph 3 under the sub heading, *‘*Loss of Otd from the LALv1 neuroblast results in lineage identity transformation.*’*

The use of *Insc-Gal4* was necessary because MARCM experiments suggested that Otd is required in the neuroblast to maintain the central complex identity of the LALv1 lineage. None of the other drivers used in the study were able to drive RNAi expression as early. We have now mentioned this in the text (see second sentence in paragraph 4 under the sub heading, ‘Determining the lineage identity of the transformed neurons’).

*2) Comparison of 4B and F shows a single cluster of otd+ cells that coincide with LALv1 in “B”, but in “F” part of the weakened otd+ cluster is now ascribed to two different clusters: part to LALv1 [yellow outline] and the remainder to ALv1 [which should not express otd?]. I am not convinced that we are seeing 2 distinct clusters rather than a split version of ALv1. It would be good to have a more convincing example of the appearance of a distinct GH146 cluster with the reduction in otd*.

In this experiment we used the *Insc-Gal4* to drive and Otd RNAi in all neuroblasts and used *GH146-QF>QUAS-Tomato::HA* to visualize the neurons. However, as RNAi is not always completely penetrant, we often did not observe a complete loss of Otd in the LALv1 lineage. The brain shown in the previous Figure 4 (current Figure 5) shows the strongest reduction in Otd in the LALv1 lineage. We have now additionally shown an *otd*^*-/-*^ MARCM clonal brain in which all four neuroblasts lineages have been recovered. Please see Figure 5.

Regarding splitting of the cell clusters, please see response 3 below.

*3) The ALv1 lineage is a potential complication because it is a ventral lineage of antennal lobe projection neurons that is situated next to LALv1 in the adult. These cells have an anatomy that is similar to the cells of the transformed LALv1 and the GH146 driver expresses in these cells. The authors should have recovered many ALv1 clones in their experiments. It is important to know how they distinguished ALv1 clones from the transformed LALv1*.

We thank the reviewers for pointing out this potential complication. The GH146 driver labels only about 6-10 cells of the ALv1 lineage. These neurons are indeed antennal lobe projection interneurons, however, their neuroanatomy is very different from the *otd*^*-/-*^ LALv1 lineage. The entry point of the axons of these two lineages into the antennal lobe is diametrically opposite to each other; while the *otd*^*-/-*^ LALv1 lineage enters the lobe medially, ALv1 (as well as the *otd*^*-/-*^ ALv1) enters it laterally (see asterisks in Figure 13). Most strikingly, their axons exit the lobe towards the protocerebrum using different tracts; while the *otd*^*-/-*^ LALv1 lineage uses the medial antennal lobe tract, the ALv1 (as well as the *otd*^*-/-*^ ALv1) use the mediolateral antennal lobe tract (see arrow in Author response 1A,B). Both these tracts are visible in Figure 5. Finally, while the *otd*^*-/-*^ LALv1 lineage first innervates the calyx of the mushroom body and then the lateral horn, the GH146-labelled neurons of ALv1 (as well as the *otd*^*-/-*^ ALv1) largely innervate only the lateral horn (see Figure 13). We have brought out these points in the main text as well. Please see paragraph 3 under the subheading, *‘*Determining the lineage identity of the transformed neurons’ (also see Figure 13).

The ALv1 lineage does not express Otd and, as expected, we do not see any gross defects in any of the *otd*^*-/-*^ ALv1 clones we have recovered (see Figure 13). Moreover, although we have recovered the *otd*^*-/-*^ LALv1 lineage along with other neuroblast clones of the antennal lobe including the ALv1 lineage, we frequently recover it with no other clones in the brain. In these clones, we observe no splitting of the cell cluster and the mutant lineage never uses the mediolateral antennal lobe tract.

Given these data we conclude that the *otd*^*-/-*^ LALv1 lineage and the *otd*^*-/-*^ ALv1 lineage have two distinct clonal origins (are two distinct neuroblast lineages) and are not a result of splitting of the *otd*^*-/-*^ LALv1 lineage.Author response image 1.WT (A) and *otd*^*-/-*^ (B) ALv1 lineage. Note the position of the cell body (yellow dotted lines), the lateral point of entry of axon tract into the antennal lobe (indicated by the asterisk), the use of the mediolateral antennal lobe tract (indicated by the arrow) and the predominant innervation of the lateral horn of both the WT (A) and the *otd*^*-/-*^ ALv1 lineage (B). Midline is represented by the yellow line. Genotype in A: *FRT19A/FRT19A,Tub-Gal80,hsFLP; GH146-Gal4,UAS-mCD8::GFP/+.* Genotype in B*: FRT19A, oc*^*2*^*/FRT19A,Tub-Gal80,hsFLP; GH146-Gal4,UAS-mCD8::GFP/+.*

*4) There are other pieces of information that would help establish the correspondence of the LALv1 cells to the LALv1otd- neurons: 1) The observation that the LEp tract is missing when the lineage is transformed is a compelling piece of information. In addition to the reconstruction [or instead of it], it would be good to show an appropriate confocal section showing the loss of the tract on the transformed side and its presence on the control side. 2) If they have them, it would be good to show images of the transformed clones in the larval stage. The position of the neuroblast and its cell cluster should be in its normal position but its axon bundle would then be taking the transformed trajectory*.

We thank the reviewers for pointing this out. We have reorganized the text and added an entire new results section under the sub heading, ‘Determining the lineage identity of the transformed neurons’ in which we systematically list out three different ways by which it is possible to infer correspondence between the LALv1 and the transformed LALv1 neurons. 1) The appearance of an extra antennal lobe lineage (see Figure 5). 2) The loss of the LEp tract (we have shown the confocal sections instead of the 3D reconstruction; see Figure 6). 3) The use of a molecular marker, Acj6 that allows the unambiguous identification of the LALv1 lineage in both the WT and mutant conditions (see Figure 7).

We have also attempted to trace the *otd*^*-/-*^ LALv1 lineage into the larval brain. In these experiments we recovered *GH146-Gal4* labelled *otd*^*-/-*^ LALv1 clones, however these clones were largely indistinguishable from the ALad1 lineage (except for the loss of Otd immunolabelling corresponding to the larval LALv1 lineage). This is because in the larval brain, the cell bodies of LALv1 and ALad1 lie closely apposed to each other and when the LALv1 lineage is transformed, they also share similar trajectories. However, in a rare clone, the *otd*^*-/-*^ LALv1 lineage retained a few projections into the loVM tract, allowing us to unambiguously identify it as the LALv1 lineage. In this brain preparation we observed that the *otd*^*-/-*^ LALv1 lineage sent a majority of its projections via the medial antennal lobe tract (see Figure 14), confirming that the *otd*^*-/-*^ LALv1 lineage indeed transforms to an ALad1 neuroblast lineage like fate.Author response image 2.Note the larval *otd*^*-/-*^ LALv1 lineage on the right hemisphere, which shows a loss of Otd immunolabelling in B. A few neurites of this lineage enter the loVM tract (cyan arrow in A,C), while most take the medial antennal lobe tract towards the protocerebrum (magenta arrow in A,C). Midline is represented by the yellow line. Genotype: *FRT19A, otd*^*YH13*^*/FRT19A,Tub-Gal80,hsFLP; GH146-Gal4,UAS-mCD8::GFP/+./+.*